# Differentially Private Clustered Federated Learning

**Saber Malekmohammadi**                     *saber.malekmohammadi@uwaterloo.ca*
*Mila - Quebec AI institute, Montreal, Canada*
*School of Computer Science, University of Waterloo, Waterloo, Canada*

**Afaf Taik** *Mila - Quebec AI institute, Montreal, Canada*

**Golnoosh Farnadi** *Mila - Quebec AI institute, Montreal, Canada*
*School of Computer Science, McGill University, Montreal, Canada*
*Université de Montréal, Montréal, Canada*

**Reviewed on OpenReview:** *https://openreview.net/forum?id=JSsko0a4yr*

## Abstract

Federated learning (`FL`), which is a decentralized machine learning (`ML`) approach, often incorporates differential privacy (`DP`) to provide rigorous data privacy guarantees to clients. Previous works attempted to address high *structured* data heterogeneity in vanilla `FL` settings through clustering clients (a.k.a clustered `FL`), but these methods remain sensitive and prone to errors, further exacerbated by the `DP` noise, making them inappropriate for differentially private `FL` (`DPFL`) under *structured* data heterogeneity. To address this gap, we propose an algorithm for differentially private clustered `FL`, which is robust to the `DP` noise in the system and identifies the underlying clients' clusters correctly. To this end, we propose to cluster clients based on both their model updates and training loss values. Furthermore, for clustering clients' model updates at the end of the first round, our proposed approach addresses the server's uncertainties by employing large batch sizes as well as Gaussian Mixture Models (`GMM`) to reduce the impact of `DP` and stochastic noise and avoid potential clustering errors. We provide theoretical analysis to justify our approach and evaluate it across diverse data distributions and privacy budgets. Our experimental results (codes: here) show effectiveness of the approach in addressing high *structured* data heterogeneity in `DPFL`.

## 1 Introduction

Federated learning (`FL`) (McMahan et al., 2017) is a collaborative `ML` paradigm, which allows multiple clients to train a shared global model without sharing their data. However, in order for `FL` algorithms to ensure rigorous privacy guarantees against data privacy attacks (Hitaj et al., 2017; Rigaki & García, 2020; Wang et al., 2019; Zhu et al., 2019; Geiping et al., 2020), they are reinforced with `DP` (Dwork et al., 2006b;a; Dwork, 2011; Dwork & Roth, 2014). This is done in the presence of a trusted server (McMahan et al., 2018; Geyer et al., 2017) and its absence (Zhao et al., 2020; Duchi et al., 2013; 2018). In the latter case and for sample-level `DP`, each client runs `DPSGD` (Abadi et al., 2016) locally and shares its noisy model updates with the server.

A key challenge in `FL` settings is ensuring an acceptable performance across clients under heterogeneous data distributions. Several existing works focus on accuracy parity across clients with a *single* common model by agnostic `FL` (Mohri et al., 2019) and client reweighting (Li et al., 2020c;b; Zhang et al., 2023). However, a single global model often fails to adapt to the data heterogeneity across clients (Dwork et al., 2012), especially when a high data heterogeneity exists. Furthermore, when using a single model and augmenting `FL` with `DP`, different subgroups of clients are unevenly affected - even with loose privacy guarantees (Farrand et al., 2020; Fioretto et al., 2022; Bagdasaryan & Shmatikov, 2019). In fact, subgroups with minority clients experience a larger drop in model utility, due to the inequitable gradient clipping in `DPSGD` (Abadi et al., 2016; Bagdasaryan & Shmatikov, 2019; Xu et al., 2021; Esipova et al., 2023). Accordingly, some works proposed to use model

personalization by multi-task learning (Smith et al., 2017; Li et al., 2021; Marfoq et al., 2021; Wu et al., 2023), transfer learning (Li & Wang, 2019; Liu et al., 2020) and clustered FL (Ghosh et al., 2020; Mansour et al., 2020; Ruan & Joe-Wong, 2021; Sattler et al., 2019; Werner et al., 2023; Briggs et al., 2020). The latter has been proposed for vanilla FL and is suitable when "structured data heterogeneity" exists across clusters of clients (as in this work): subsets of clients can be naturally grouped together based on their data distributions and one model is learned for each group (cluster). However, as discussed by Werner et al. (2023), the existing non-private clustered FL approaches are vulnerable to errors in clustering due to their sensitivity to: 1. model initialization 2. randomness in clients' model updates due to stochastic noise. The DP noise existing in DPFL systems' training mechanism exacerbates this vulnerability by injecting more randomness.

To address the aforementioned gap, we propose a sample-level differentially private clustered FL algorithm (Algorithm 1), which uses both clients' model updates and loss values for clustering clients, making it more robust to DP/stochastic noise: 1) Justified by our theoretical analysis (Lemma 4.1 and 4.2) and in order to cluster clients correctly, our proposed algorithm uses a full batch size in the first FL round and a small batch size in the subsequent rounds, to reduce the noise in clients' model updates at the end of the first round. 2) Then, the server soft clusters clients based on these less noisy model updates using a Gaussian Mixture Model (GMM). Depending on the "confidence" of the learned GMM, the server keeps using it to soft cluster clients during the next few rounds (Section 4.3). 3) Finally, the server switches the clustering strategy to local clustering of clients based on their train loss/accuracy values in the remaining rounds. These altogether make our DP clustered FL algorithm effective and robust. The highlights of our contributions are as follows:

- We propose a sample-level DP clustered FL algorithm (R-DPCFL), which combines information from both clients' model updates and their loss values. The algorithm is robust and achieves high-quality clustering of clients, even in the presence of DP noise in the system (Algorithm 1).

- We theoretically prove that increasing clients' batch sizes in the first round (and decreasing them in the subsequent rounds) improves the server's ability to cluster clients based on their model updates at the end of the first round (Lemma 4.2).

- We show that utilizing sufficiently large batch sizes in the first round (and sufficiently small batch sizes in the next rounds) enables super-linear convergence rate for learning a GMM by the server on the clients' model updates at the end of the first round. This leads to soft clustering of clients using a GMM with a low computational overhead (Theorem 4.3).

- We extensively evaluate the proposed algorithm across diverse datasets and scenarios and demonstrate the effectiveness of our robust DP clustered FL algorithm in detecting the underlying cluster structure of clients, which leads to an overall utility improvement for the system (Section 6).

## 2 Related work

Model personalization is a technique for improving utility under data heterogeneity (Li et al., 2021; Liu et al., 2022a), which usually leverages extra computations, e.g., extra local iterations (Li et al., 2021). On the other hand, clustered FL has been proposed for personalized FL under a high *structured* data heterogeneity, where clients can be naturally partitioned into clusters: clients in the same cluster have similar data distributions, while there is a significant heterogeneity across various clusters. Existing clustered FL algorithms group clients based on their loss values (Ghosh et al., 2020; Mansour et al., 2020; Ruan & Joe-Wong, 2021; Dwork et al., 2012; Liu et al., 2022b) or their model updates (based on e.g., their euclidean distance (Werner et al., 2023; Briggs et al., 2020) or cosine similarity (Sattler et al., 2019)). As shown by Werner et al. (2023), the algorithms are prone to clustering errors in the early rounds of FL training –due to gradient stochasticity, model initialization or the form of loss functions far from their optima– which can even propagate in the subsequent rounds. This vulnerability is exacerbated in DPFL systems, due to the existing extra DP noise. Without addressing this vulnerability, Luo et al. (2024) proposed a DP clustered FL algorithm with a limited applicability, which clusters clients based on the labels that they do not have in their local data. In contrast, our DP clustered FL algorithm is applicable to any setting characterized by a number of clients, *where each client holds many data samples and needs sample-level privacy protection.* Cross-silo FL systems can be

considered as an instance, where there are fewer clients but each hold many data subjects that require protection. For example, when hospitals/banks/schools wish to federate patient/customer/student records, it is the people owning those records rather than the participating silos that should be protected. The closest study to this setting has been done by Liu et al. (2022a), which considers silo-specific sample-level DP for cross-silo FL and studies the interplay between privacy and data heterogeneity. More specifically, they show that *when clients have large datasets and "moderate" data heterogeneity exits across them*: 1. participation in FL by clients is encouraged over local training, as the FL model averaging on the server diminishes the effect of DP noise 2. under the same total privacy budget, model personalization - through mean regularized multi-task learning (MR-MTL) - leads to a better performance compared to training a single global model or local training by clients (see Appendix C.4 about MR-MTL formulation). Complementing the work, we show that model personalization (with MR-MTL), local training and even loss-based clustered FL are not efficient for DPFL with structured data heterogeneity. Accordingly, we propose our new DP clustered FL algorithm.

## 3 Definitions, notations and assumptions

There are multiple definitions of DP. We adopt the following definition to be satisfied by every client:

**Definition 3.1** (($\epsilon, \delta$)-DP (Dwork et al., 2006a)). *A randomized mechanism $\mathcal{M} : \mathcal{A} \to \mathcal{R}$ with domain $\mathcal{A}$ and range $\mathcal{R}$ satisfies ($\epsilon, \delta$)-DP if for any two adjacent inputs $\mathcal{D}, \mathcal{D}' \in \mathcal{A}$, which differ by only a single record (by replacement), and for any measurable subset of outputs $\mathcal{S} \subseteq \mathcal{R}$ it holds that*

$$Pr[\mathcal{M}(\mathcal{D}) \in \mathcal{S}] \le e^\epsilon Pr[\mathcal{M}(\mathcal{D}') \in \mathcal{S}] + \delta.$$

The gaussian mechanism randomizes the output of a query $f$ as $\mathcal{M}(\mathcal{D}) \triangleq f(\mathcal{D}) + \mathcal{N}(0, \sigma^2)$. The randomized output satisfies ($\epsilon, \delta$)-DP for a continuum of pairs ($\epsilon, \delta$): for all $\epsilon, \delta \in (0, 1)$ and $\sigma > \frac{\sqrt{2 \ln(1.25/\delta)}}{\epsilon} \Delta_2 f$, where $\Delta_2 f \triangleq \max_{\mathcal{D}, \mathcal{D}'} \| f(\mathcal{D}) - f(\mathcal{D}') \|_2$ is the $l_2$-sensitivity of the query $f$ with respect to its input. Also, the $\epsilon$ and $\delta$ privacy parameters resulting from running Gaussian mechanism depend on the quantity $z = \frac{\sigma}{\Delta_2 f}$ (called "noise scale"). We consider a DPFL system (see Figure 2, left), where there are $n$ clients running DPSGD with the same "sample-level" privacy parameters ($\epsilon, \delta$): the set of information (including model updates and cluster selections) sent by client $i$ to the server satisfies ($\epsilon, \delta$)-DP for all adjacent datasets $\mathcal{D}_i$ and $\mathcal{D}'_i$ of the client $i$ differing in one sample (by replacement).

Let $x \in \mathcal{X} \subseteq \mathbb{R}^d$ and $y \in \mathcal{Y} = \{1, \ldots, C\}$ denote an input data point and its target label. Client $i$ holds dataset $\mathcal{D}_i$ with $N_i$ samples from distribution $P_i(x, y) = P_i(y|x)P_i(x)$. Let $h : \mathcal{X} \times \boldsymbol{\theta} \to \mathbb{R}^C$ be the predictor function, which is parameterized by $\boldsymbol{\theta} \in \mathbb{R}^p$. Also, let $\ell : \mathbb{R}^C \times \mathcal{Y} \to \mathbb{R}_+$ be the used loss function (cross-entropy loss). Client $i$ in the system has empirical train loss $f_i(\boldsymbol{\theta}) = \frac{1}{N_i} \sum_{(x,y) \in \mathcal{D}_i} [\ell(h(x, \boldsymbol{\theta}), y)]$, with minimum value $f_i^*$. There are $E$ communication rounds indexed by $e$ and $K$ local epochs with learning rate $\eta_l$ during each round. There are $M$ clusters of clients indexed by $m$, and the server holds $M$ cluster models $\{\boldsymbol{\theta}_m^e\}_{m=1}^M$ for them at the beginning of round $e$. The value of $M$ maybe unknown at the beginning. Clients $i$ and $j$ belonging to the same cluster have the same data distributions $P_i(x, y) = P_j(x, y)$, while there is a high data heterogeneity across clusters. $s(i)$ denotes the true cluster of client $i$ and $R^e(i)$ denotes the cluster assigned to it at the beginning of round $e$. Let us assume the batch size used by client $i$ in the first round $e = 1$ is $b_i^1$, which may differ from the batch size $b_i^{>1}$ that it uses in the rest of the rounds $e > 1$. At the $t$-th gradient update during the round $e$, client $i$ uses batch $\mathcal{B}_i^{e,t}$ with size $b_i^e$, and computes the following DP noisy batch gradient:

$$\tilde{g}_i^{e,t}(\boldsymbol{\theta}) = \frac{1}{b_i^e} \left[ \left( \sum_{j \in \mathcal{B}_i^{e,t}} \bar{g}_{ij}(\boldsymbol{\theta}) \right) + \mathcal{N}(0, \sigma_{i,\text{DP}}^2 \mathbb{I}_p) \right], \tag{1}$$

where $\bar{g}_{ij}(\boldsymbol{\theta}) = \texttt{clip}(\nabla \ell(h(x_{ij}, \boldsymbol{\theta}), y_{ij}), c)$, and $c$ is a clipping threshold to clip sample gradients: for a given vector $\mathbf{v}$, $\texttt{clip}(\mathbf{v}, c) = \min\{\|\mathbf{v}\|, c\} \cdot \frac{\mathbf{v}}{\|\mathbf{v}\|}$. Also, $\mathcal{N}$ is the Gaussian noise distribution with variance $\sigma_{i,\text{DP}}^2$, where $\sigma_{i,\text{DP}} = c \cdot z_i(\epsilon, \delta, b_i^1, b_i^{>1}, N_i, K, E)$, and $z_i$ is the noise scale needed for achieving ($\epsilon, \delta$)−DP by client $i$, which can be determined with a privacy accountant, e.g., Rényi-DP accountant (Mironov et al., 2019) used in this work, which is capable of accounting composition of *heterogeneous* DP mechanisms (Mironov, 2017). The privacy parameter $\delta$ is fixed to $10^{-4}$ in this work with $\delta < N_i^{-1}$ for every client $i$. For a random

$\mathbf{v} = (v_1, \ldots, v_p)^\top \in \mathbb{R}^{p \times 1}$, we define $\mathtt{Var}(\mathbf{v}) := \sum_{j=1}^{p} \mathbb{E}[(v_j - \mathbb{E}[v_j])^2]$, i.e., the sum of the variances of all its elements. Table 1 in the appendix summarizes the used notations. Finally, we have the following assumption:

**Assumption 3.2.** *The stochastic gradient $g_i^{e,t}(\boldsymbol{\theta}) = \frac{1}{b_i^e} \sum_{j \in \mathcal{B}_i^{e,t}} g_{ij}(\boldsymbol{\theta})$ is an unbiased estimator of $\nabla f_i(\boldsymbol{\theta})$ with a bounded variance: $\forall \boldsymbol{\theta} \in \mathbb{R}^p : \mathtt{Var}(g_i^{e,t}(\boldsymbol{\theta})) \leq \sigma_{i,g}^2(b_i^e)$. The tight bound $\sigma_{i,g}^2(b_i^e)$ is a constant depending only on the used batch size $b_i^e$: the larger $b_i^e$, the smaller $\sigma_{i,g}^2(b_i^e)$.*

## 4 Methodology and proposed algorithm

As discussed by Werner et al. (2023), clustered `FL` algorithms which cluster clients based on their loss values (Mansour et al., 2020; Ghosh et al., 2020; Ruan & Joe-Wong, 2021), i.e., assign client $i$ to cluster $R^e(i) = \arg\min_m f_i(\boldsymbol{\theta}_m^e)$ at the beginning of round $e$, are prone to clustering errors in the first few rounds, mainly due to random initialization of cluster models $\{\boldsymbol{\theta}_m^0\}_{m=1}^M$. On the other hand, clustering clients based on their model updates (gradients) (Werner et al., 2023; Briggs et al., 2020; Sattler et al., 2019) makes sense only when the updates are obtained on the same model initialization. Additionally, even if we assume these algorithms can initially cluster clients perfectly in each round $e$, the clients' model updates (gradients) will approach zero as the clusters' model parameters converge. Hence, clients from different clusters may appear to belong to the same cluster, which again results in clustering mistakes.

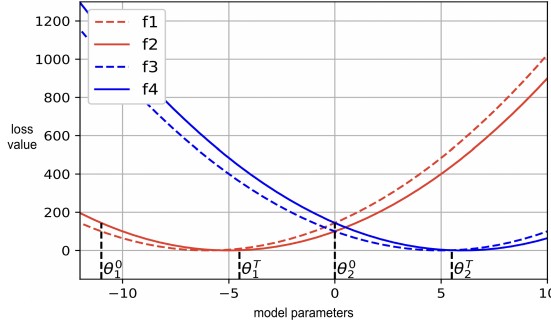

Figure 1: Loss-based clustering algorithms miscluster in the initial rounds, due to model initialization. Also, even with the assumption of perfect clustering of clients in the first rounds, clustering algorithms based on gradients (model updates) leads to clustering errors in the last rounds, due to the gradients approaching zero.

We now elaborate on why clustering clients based on their losses (model updates) is prone to errors in the first (last) rounds with an example. Consider Figure 1, where there are $M = 2$ clusters (red and blue) and $n = 4$ clients. The clients in the red cluster have loss functions $f_1(\theta) = 4(\theta + 6)^2$ and $f_2(\theta) = 4(\theta + 5)^2$ with optimum cluster parameter $\theta_1^\infty = -5.5$. Also, the the clients in the blue cluster have loss functions $f_3(\theta) = 4(\theta - 5)^2$ and $f_4(\theta) = 4(\theta - 6)^2$ with optimum cluster parameter $\theta_2^\infty = 5.5$. Clustering algorithms, which cluster clients based on their loss values on clusters' models, are vulnerable to model initialization. For example, if we initialize the clusters' parameters with $\theta_1^0 = -11$ and $\theta_2^0 = 0$ (shown in the figure), all four clients will initially select cluster 2, since they have smaller losses on its parameter. At $\theta_2^0 = 0$, the average of clients' gradients (model updates) is zero, so all clients will remain stuck at $\theta_2^0$ and will always select cluster 2.

On the other hand, clustering clients based on their model updates (gradients) (Werner et al., 2023; Briggs et al., 2020; Sattler et al., 2019) have clearly issues. One of these issues appears after some rounds of training. For instance, even if we assume these algorithms can initially cluster clients "perfectly" in each round $e$, the clients' model updates (gradients) will approach zero as the clusters' models converge to their optimum parameters. Hence, clients from different clusters may appear to belong to the same cluster, which results in clustering mistakes. For example, as shown in Figure 1, let us assume after $T$ rounds of "correct" clustering of clients, the clusters' parameters get to $\theta_1^T = -4.5$ and $\theta_2^T = 5.5$. At these parameters, clients 1 and 2 (which have been "correctly" assigned to cluster 1 so far) will have gradients $f_1'(\theta_1^T) = 12$ and $f_2'(\theta_1^T) = 4$. Similarly, clients 3 and 4 (which have been "correctly" assigned to cluster 2 so far) will have $f_3'(\theta_2^T) = 4$ and $f_4'(\theta_2^T) = -4$. We see that $f_2'$ is closer to $f_3'$ and $f_4'$ than to $f_1'$, and in the next round it will be wrongly

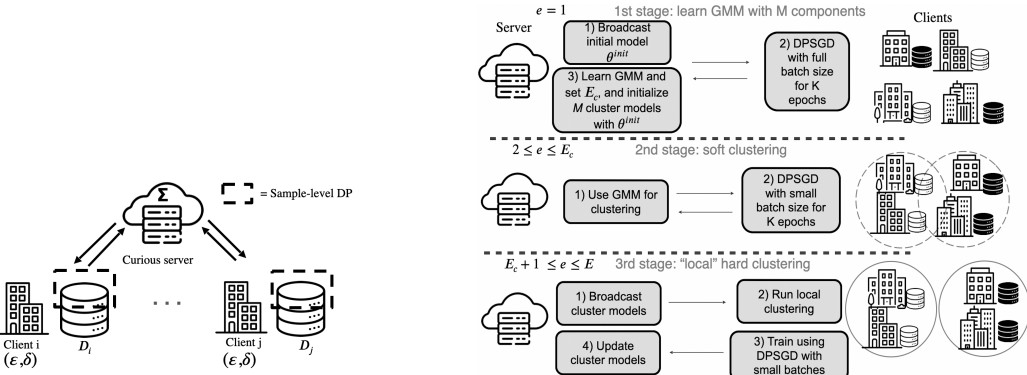

Figure 2: **Left:** Considered threat model in this work, where client $i$ has local train data $\mathcal{D}_i$ and "sample-level" DP privacy parameters $(\epsilon, \delta)$, and does not trust any external party. **Right:** Three main stages of R-DPCFL.

assigned to cluster 2 (with clients 3 and 4). This happens while the clients are clearly distinguishable based on their losses, *as some progress in training has been made after $T$ rounds*: $f_2(\theta_1^T) = 1$, while $f_2(\theta_2^T) = 21^2$, which clearly means that client 2 correctly belongs to cluster 1. Therefore, after making some progress in training the clusters' models, it makes more sense to use a loss-based clustering strategy than using a strategy based on clients' gradients (model updates).

Motivated by this vulnerability, which will get exacerbated by DP noise, we next propose a DP clustered FL algorithm which starts with clustering clients based on their model updates for the first several rounds and then switches its strategy to cluster clients based on their loss values. We augment this idea with some other non-obvious techniques to enhance the clustering accuracy.

### 4.1 R-DPCFL **algorithm**

Our proposed R-DPCFL algorithm has three main steps (also see Figure 2, right and Algorithm 1):

1. In the first round, clients train the initial model $\boldsymbol{\theta}^{init}$ locally. They use full batch sizes in this round to make their model updates $\{\Delta\tilde{\boldsymbol{\theta}}_i^1\}_{i=1}^n$ less noisy. **Note that even when clients have a limited memory budget, they can still perform DPSGD with full batch size and no computational overhead by using gradient accumulation technique (see Appendix G)**. Then, the server soft clusters them by learning GMM on their model updates. The number of clusters (**M**) is either given or can be found by maximizing the confidence of the learned GMM (Section 4.3).

2. During the subsequent rounds $e \in \{2, \ldots, E_c\}$, the server uses the learned GMM to soft-cluster clients: client $i$ uses a small batch size $b_i^{>1}$ and contributes to the training of each cluster ($m$) model proportional to the probability of its assignment to that cluster ($\pi_{i,m}$). The number of rounds $E_c$ is set based on "confidence level" of the learned GMM (Section 4.3).

3. After some progress during the first $E_c$ rounds, clients' train loss/accuracy values on cluster models $\{\boldsymbol{\theta}_m^{E_c}\}_{m=1}^M$ are meaningful. Accordingly, clients use them locally during the remaining rounds to *privately* select a cluster using exponential mechanism (with parameter $\epsilon_{\text{select}}$, Equation (8)).

In Sections 4.2 and 4.4, we provide theoretical justifications for the usage of full batch size in the first round.

### 4.2 **Reducing** GMM **uncertainty via using full (small) batch sizes in the first (next) round(s)**

It is the DP noise in $\{\Delta\tilde{\boldsymbol{\theta}}_i^1\}_{i=1}^n$ that makes it hard for the server to cluster clients by learning a GMM on the model updates. Lemma 4.1 extends a similar result in (Malekmohammadi et al., 2024) and shows that the amount of noise in $\Delta\tilde{\boldsymbol{\theta}}_i^e$ at the end of each round $e$ rapidly declines when $b_i^e$ increases.

---

**Algorithm 1:** R-DPCFL

---

**Input:** Initial parameter $\boldsymbol{\theta}^{init}$, dataset sizes $\{N_1, \ldots, N_n\}$, batch sizes $\{b_1^{>1}, \ldots, b_n^{>1}\}$, clip bound $c$, local epochs $K$, global round $E$, number of clusters $M$ (optional)

**Output:** cluster models $\{\boldsymbol{\theta}_m^E\}_{m=1}^M$

1   **for** *each client $i \in \{1, \ldots, n\}$* **do**
2      $b_i^1 \leftarrow N_i$ ;              `// full batch size`
3      $z_i \leftarrow \mathtt{RDP}(\epsilon, \delta, b_i^1, b_i^{>1}, N_i, K, E)$

4   **for** $e \in \{1, \ldots, E\}$ **do**
5      **if** $e = 1$ **then**
6          **for** *each client $i \in \{1, \ldots, n\}$ **in parallel** do*
7              $\Delta\tilde{\boldsymbol{\theta}}_i^1 \leftarrow \mathtt{DPSGD}\,(\boldsymbol{\theta}^{init}, b_i^1, N_i, K, z_i, c)$
8          on server:
9          **if** $M$ *is unknown* **then**
10              $M = \arg\max_{M'} \mathtt{MSS}\Big(\mathtt{GMM}(\Delta\tilde{\boldsymbol{\theta}}_1^1, \ldots, \Delta\tilde{\boldsymbol{\theta}}_n^1; M')\Big)$ ;      `// set M (Section 4.3)`
11          $\{\pi_1, \ldots, \pi_n, \mathtt{MPO}\} = \mathtt{GMM}(\Delta\tilde{\boldsymbol{\theta}}_1^1, \ldots, \Delta\tilde{\boldsymbol{\theta}}_n^1; M)$ ;      `// 1st stage: GMM`
12          set $E_c(\mathtt{MPO})$ ;              `// set E_c (Section 4.3)`
13          Initialize cluster models uniformly: $\boldsymbol{\theta}_1^2 = \ldots = \boldsymbol{\theta}_M^2 = \boldsymbol{\theta}^{init}$
14          continue ;              `// go to round e = 2`
15      **else if** $e \in \{2, \ldots, E_c\}$ **then**
16          **for** *each client $i \in \{1, \ldots, n\}$* **do**
17              $R^e(i) \leftarrow m$ *with probability* $\pi_i[m]$ ;      `// 2nd stage: soft clustering`
18      **else**
19          on server: broadcast cluster models $\{\boldsymbol{\theta}_m^e\}_{m=1}^M$ to all clients to run clustering *locally*
20          **for** *each client $i \in \{1, \ldots, n\}$* **do**
21              $R^e(i) = \arg\min_m f_i(\boldsymbol{\theta}_m^e)$ ;    `// 3rd stage: `*private*` loss/accuracy-based clustering`
22      **for** *each client $i \in \{1, .., n\}$ **in parallel** do*
23          $\Delta\tilde{\boldsymbol{\theta}}_i^e \leftarrow \mathtt{DPSGD}\,(\boldsymbol{\theta}_{R^e(i)}^e, b_i^{>1}, N_i, K, z_i, c)$ ;      `// batch size `$b_i^{>1}$
24      on server:
25      **for** *each client $i \in \{1, \ldots, n\}$* **do**
26          $w_i^e \leftarrow \frac{1}{\sum_{j=1}^n \mathbb{1}_{R^e(j)=R^e(i)}}$
27      **for** $m \in \{1, \ldots, M\}$ **do**
28          $\boldsymbol{\theta}_m^{e+1} \leftarrow \boldsymbol{\theta}_m^e + \sum_{i \in \{1, \ldots, n\}} \mathbb{1}_{R^e(i)=m} w_i^e \Delta\tilde{\boldsymbol{\theta}}_i^e$

---

**Lemma 4.1.** *Let us assume $\boldsymbol{\theta}_i^{e,0}$ is the starting model parameter for client $i$ at the beginning of round $e$. After $K$ local epochs with step size $\eta_l$, the client generates the noisy* DP *model update $\Delta\tilde{\boldsymbol{\theta}}_i^e(b_i^e)$ at the end of the round. The amount of noise in the resulting model update can be quantified as:*

$$(\sigma_i^e(b_i^e))^2 := \mathtt{Var}(\Delta\tilde{\boldsymbol{\theta}}_i^e(b_i^e)|\boldsymbol{\theta}_i^{e,0}) \approx K \cdot N_i \cdot \eta_l^2 \cdot \frac{pc^2 z_i^2(\epsilon, \delta, b_i^1, b_i^{>1}, N_i, K, E)}{(b_i^e)^3}. \tag{2}$$

The first conclusion from the lemma is that the noise level in $\Delta\tilde{\boldsymbol{\theta}}_i^e$ rapidly declines as $b_i^e$ increases: See Figure 3 and the effect of batch size $b_i^1$ on $\mathtt{Var}(\Delta\tilde{\boldsymbol{\theta}}_i^1|\boldsymbol{\theta}^{init})$ (on the left); and the effect of batch size $b_i^{>1}$ on $\mathtt{Var}(\Delta\tilde{\boldsymbol{\theta}}_i^e|\boldsymbol{\theta}_i^{e,0})$ $(e > 1)$ (on the right). Let us consider $e = 1$ especially: If all clients use full batch sizes in the first round (i.e., $b_i^1 = N_i$ for every client $i$), it becomes much easier for the server to cluster them at the end of the first round by learning a GMM on $\{\Delta\tilde{\boldsymbol{\theta}}_i^1\}_{i=1}^n$, as their updates become more separable. An illustration of this is shown in Figure 5. As the second key takeaway, Figure 3 left shows that **in order to make** $\{\Delta\tilde{\boldsymbol{\theta}}_i^1\}_{i=1}^n$

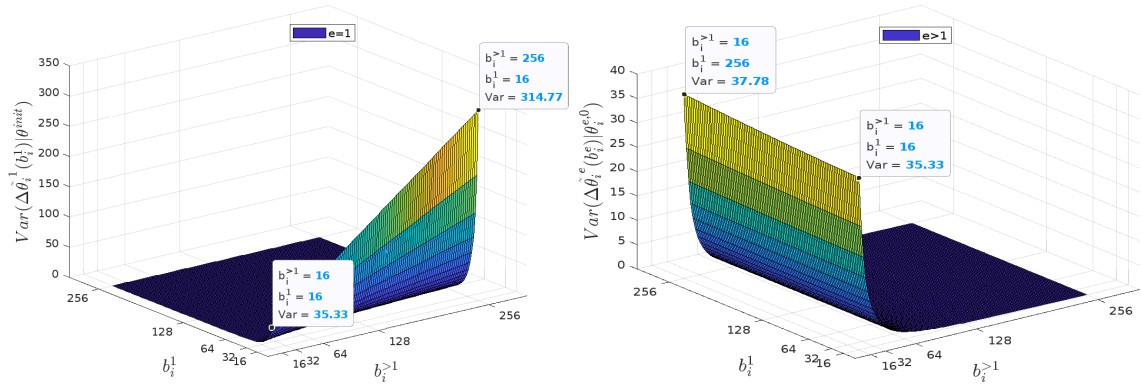

Figure 3: Plot of $\text{Var}(\Delta\tilde{\boldsymbol{\theta}}_i^1(b_i^1)|\boldsymbol{\theta}_i^{init})$ (**left**) and $\text{Var}(\Delta\tilde{\boldsymbol{\theta}}_i^e(b_i^e)|\boldsymbol{\theta}_i^{e,0})$ ($e > 1$) (**right**) v.s. both $b_i^1$ and $b_i^{>1}$. There are two clear takeaways: 1) for all $e \in \{1, \cdots, E\}$, $\text{Var}(\Delta\tilde{\boldsymbol{\theta}}_i^e(b_i^e)|\boldsymbol{\theta}_i^{e,0})$ decreases with $b_i^e$ steeply (from Lemma 4.1). 2) The effect of $b_i^{>1}$ on $\text{Var}(\Delta\tilde{\boldsymbol{\theta}}_i^1(b_i^1)|\boldsymbol{\theta}_i^{init})$ (left figure) is considerable (see Figure 4 for the plot of $z_i(\epsilon, \delta, b_i^1, b_i^{>1}, N_i, K, E)$ v.s. $b_i^1$ and $b_i^{>1}$). The results are obtained on CIFAR10 from Rényi-DP accountant (Mironov et al., 2019) in a setting with $N_i = 6600, \epsilon = 5, \delta = 10^{-4}, c = 3, K = 1, E = 200, p = 11,181,642, \eta_l = 5 \times 10^{-4}$.

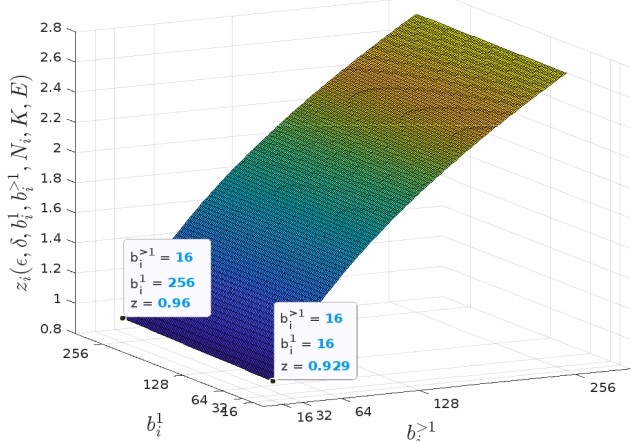

Figure 4: Plot of $z_i(\epsilon, \delta, b_i^1, b_i^{>1}, N_i, K, E)$ v.s. $b_i^1$ and $b_i^{>1}$ obtained from Rényi-DP Accountant (Mironov et al., 2019) in a setting with $N_i = 6600, \epsilon = 5, \delta = 10^{-4}, K = 1, E = 200$. As observed, $z_i$ is a sublinearly increasing function of both $b_i^1$ and $b_i^{>1}$. Also, the effect of $b_i^{>1}$ is much more than the effect of $b_i^1$. The reason is that $b_i^{>1}$ is used in $E - 1$ rounds, while $b_i^1$ is used only in the first round. So the value of $b_i^{>1}$ affects $z_i$ the most.

less noisy, we have to make $\{b_i^1\}_{i=1}^n$ as large as possible and also keep $\{b_i^{>1}\}_{i=1}^n$ small[1]. In the next section, we will provide a theoretical justification for the observation in Figure 5.

### 4.2.1 Effect of batch sizes $\{b_i^1\}_{i=1}^n$ on the separability of clusters

In order to theoretically understand the reason behind the observation in Figure 5, let us assume clients have the same dataset sizes and first batch sizes for simplicity: $\forall i : N_i = N, b_i^1 = b^1$. Also, remember that $\boldsymbol{\theta}_i^{1,0} = \boldsymbol{\theta}^{init}$. Having uniform privacy parameters $(\epsilon, \delta)$, we have: $\forall i : (\sigma_i^1(b^1))^2 := \text{Var}[\Delta\tilde{\boldsymbol{\theta}}_i^1(b^1)|\boldsymbol{\theta}^{init}] = (\sigma^1(b^1))^2$. Hence, we can consider the model updates $\{\Delta\tilde{\boldsymbol{\theta}}_i^1(b^1)\}_{i=1}^n$ as the samples from a mixture of $M$ Gaussians with mean, covariance matrix, prior probability parameters: $\psi^*(b^1) = \{\mu_m^*(b^1), \Sigma_m^*(b^1), \alpha_m^*\}_{m=1}^M$, where $\forall m : \alpha_m^* > 0$ and $\mu_m^*(b^1) \neq \mu_{m'}^*(b^1)$ $(m \neq m')$ (due to data heterogeneity across clusters):

---

[1]There is a close relation between the result of Lemma 4.1 and the law of large numbers. See Appendix F for more details

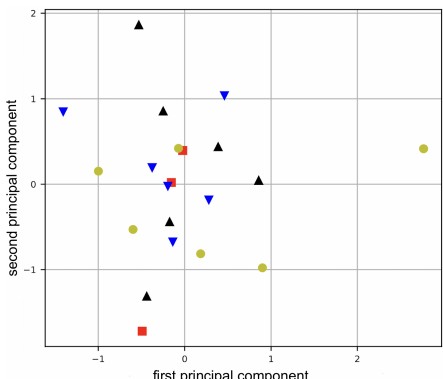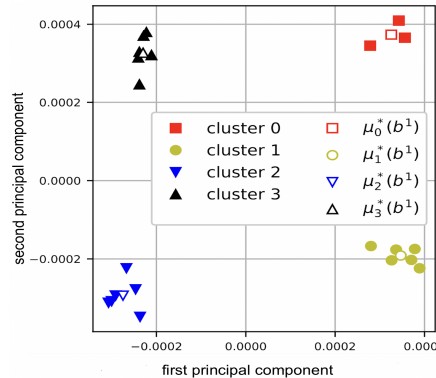

Figure 5: 2D PCA visualization of updates $\{\Delta\tilde{\boldsymbol{\theta}}_i^1\}_{i=1}^n$. **Left:** $\epsilon_i = 10$, $b_i^e = 32$ for all $i$ and $e$. **Right:** $\epsilon_i = 10$, $b_i^1 = b^1 = N = 6600$, **i.e., full batch sizes** (assuming $N_i = N = 6600$ for all clients), and $b_i^{>1} = 32$ for all $i$. The empty markers show the centers of the Gaussian components. The model updates are obtained from clients running DPSGD for $K = 1$ epochs locally on CIFAR10 with covariate shift (rotation) across clusters.

$$\mu_m^*(b^1) := \mathbb{E}\left[\Delta\tilde{\boldsymbol{\theta}}_i^1(b_i^1)\middle|\boldsymbol{\theta}^{init}, b_i^1 = b^1, s(i) = m\right], \tag{3}$$

$$\Sigma_m^*(b^1) := \mathbb{E}\left[\left(\Delta\tilde{\boldsymbol{\theta}}_i^1(b_i^1) - \mu_m^*(b^1)\right)\left(\Delta\tilde{\boldsymbol{\theta}}_i^1(b_i^1) - \mu_m^*(b^1)\right)^\top\middle|\boldsymbol{\theta}^{init}, b_i^1 = b^1, s(i) = m\right] = \frac{(\sigma^1(b^1))^2}{p}\mathbb{I}_p, \tag{4}$$

where the last equality is from $\mathtt{Var}[\Delta\tilde{\boldsymbol{\theta}}_i^1|\boldsymbol{\theta}^{init}, b_i^1 = b^1] = \mathbb{E}[\|\Delta\tilde{\boldsymbol{\theta}}_i^1 - \mu_{s(i)}^*(b^1)\|^2] = (\sigma^1(b^1))^2$ and that the noises existing in each of the $p$ elements of $\Delta\tilde{\boldsymbol{\theta}}_i^1$ are *i.i.d* (hence, $\Sigma_m^*(b^1)$ is a diagonal covariance matrix with equal diagonal elements). Intuitively, we expect more separation between the true Gaussian components $\{\mathcal{N}(\mu_m^*(b^1), \Sigma_m^*(b^1))\}_{m=1}^M$, from which clients' updates $\{\Delta\tilde{\boldsymbol{\theta}}_i^1\}_{i=1}^n$ are sampled, to make the model updates more distinguishable for the server. Next, we show that the overlap between the Gaussian components $\{\mathcal{N}(\mu_m^*(b^1), \Sigma_m^*(b^1))\}_{m=1}^M$ decreases *fast* with $b^1$:

**Lemma 4.2.** *Let* $\Delta_{m,m'}(b^1) := \|\mu_m^*(b^1) - \mu_{m'}^*(b^1)\|$ *when* $\forall i : b_i^1 = b^1$. *The overlap between components* $\mathcal{N}(\mu_m^*(b^1), \Sigma_m^*(b^1))$ *and* $\mathcal{N}(\mu_{m'}^*(b^1), \Sigma_{m'}^*(b^1))$ *is* $O_{m,m'} = 2Q(\frac{\sqrt{p}\Delta_{m,m'}(b^1)}{2\sigma^1(b^1)})$, *where* $(\sigma^1(b^1))^2 := \mathtt{Var}[\Delta\tilde{\boldsymbol{\theta}}_i^1|\boldsymbol{\theta}^{init}, b_i^1 = b^1]$ *and* $Q(\cdot)$ *is the Q function. Furthermore, if we increase* $b_i^1 = b^1$ *to* $b_i^1 = kb^1 \leq N$ *(for all* $i$*), we have* $O_{m,m'} \leq 2Q(\frac{\sqrt{kp}\Delta_{m,m'}(b^1)}{2\rho\sigma^1(b^1)})$, *where* $1 \leq \rho \in \mathcal{O}(1)$ *is a small constant.*

The terms $\Delta_{m,m'}(b^1)$ and $\sigma^1(b^1)$ represent the "data heterogeneity level across clusters $m$ and $m'$" and "privacy sensitivity of their clients", respectively. We define their "separation score" as $\mathtt{SS}(m, m') := \frac{\sqrt{p}\Delta_{m,m'}(b^1)}{2\sigma^1(b^1)} = \frac{\Delta_{m,m'}(b^1)}{2\sigma^1(b^1)/\sqrt{p}}$. The larger $\mathtt{SS}(m, m')$, the smaller overlap $O_{m,m'} = 2Q(\mathtt{SS}(m, m'))$. Based on the form of the Q function, an $\mathtt{SS}(m, m')$ above 3 can be considered as a complete separation of the components.

### 4.3 Tuning hyper-parameters of R-DPCFL

As we observed in Lemma 4.2, the separation score $\mathtt{SS}(m, m')$ (the overlap $O_{m,m'}$) increases (decreases) as $b^1$ increases. Remember that $\mathtt{SS}(m, m') = \frac{\Delta_{m,m'}(b^1)}{2\sigma^1(b^1)/\sqrt{p}}$, and note that $(\sigma^1(b^1))^2/p$ is the value of diagonal elements of covariance matrices of Gaussian components, which the GMM aims to learn (see Equation (4)). Therefore, when the GMM is learned, we can use its parameters to get an estimate score $\hat{\mathtt{SS}}(m, m')$ for every pair of clusters $m$ and $m'$. Then, we can define the "minimum pairwise separation score" as $\mathtt{MSS}(\epsilon, \delta, \{b_i^1\}_{i=1}^n, \{b_i^{>1}\}_{i=1}^n) = \min_{m,m'} \hat{\mathtt{SS}}(m, m') \in [0, +\infty)$ as a **measure of confidence** of the learned GMM in its identified clusters. The larger the MSS of a learned GMM, the more "confident" it is in its clustering decisions. For instance, if we learn

a GMM on Figure 5 left, it will have a much smaller MSS than when we learn a GMM on Figure 5 right. We can similarly define the estimated "maximum pairwise overlap" for a learned GMM as MPO $= 2Q(\text{MSS}) \in [0, 1)$, as a **measure of uncertainty** of the learned GMM. As we will explain, we can use MSS and MPO of the learned GMM to set the hyper-parameters of R-DPCFL, namely $b_i^{>1}$, $E_c$ and $M$ (when the number of clusters is unknown).

### 4.4 Convergence rate of EM for learning GMM

Let us define the maximum pairwise overlap between the components in $\psi^*(b^1) = \{\mu_m^*(b^1), \Sigma_m^*(b^1), \alpha_m^*\}_{m=1}^M$, as $O^{\max}(\psi^*(b^1)) = \max_{m,m'} O_{m,m'}(\psi^*(b^1))$. According to Lemma 4.2, when $b^1$ is large enough, $O^{\max}(\psi^*(b^1))$ decreases (like in Figure 5, right) and we can expect EM to converge to the true GMM parameters $\psi^*(b^1)$. Next, we analyze the local convergence rate of EM around $\psi^*(b^1)$.

**Theorem 4.3.** *(Ma et al., 2000) Given model updates $\{\Delta\tilde{\boldsymbol{\theta}}_i^1(b^1)\}_{i=1}^n$, as samples from a true mixture of Gaussians $\psi^*(b^1) = \{\mathcal{N}(\mu_m^*(b^1), \Sigma_m^*(b^1)), \alpha_m^*\}_{m=1}^M$, if $O^{max}(\psi^*(b^1))$ is small enough, then:*

$$\lim_{r\to\infty} \frac{\|\psi^{r+1} - \psi^*(b^1)\|}{\|\psi^r - \psi^*(b^1)\|} = o\left(\left[O^{max}(\psi^*(b^1))\right]^{0.5-\gamma}\right), \tag{5}$$

*as $n$ increases. $\psi^r$ is the GMM parameters returned by EM after $r$ iterations. $\gamma$ is an arbitrary small positive number, and $o(x)$ means it is a higher order infinitesimal as $x \to 0 : \lim_{x\to 0} \frac{o(x)}{x} = 0$.*

This means that convergence rate of EM around the true solution $\psi^*(b^1)$ is faster than how $\left[O^{\max}(\psi^*(b^1))\right]^{0.5-\gamma}$ decreases with $b^1$ (from Lemma 4.2). *Hence, as an important consequence, the computational complexity of learning the GMM in the first round also decreases fast as $b^1$ increases.*

## 5 Formal privacy guarantee of R-DPCFL

The privacy guarantee of R-DPCFL for each client $i$ in the system comes from the fact that the client runs DPSGD with a fixed DP noise variance $\sigma_{i,\text{DP}}^2 = c^2 \cdot z_i^2(\epsilon, \delta, b_i^1, b_i^{>1}, N_i, K, E)$ for all of its batch gradient computations. In the following theorem, we provide a formal privacy guarantee for the algorithm to show the sample-level privacy guarantees provided to each client $i$. We defer the proof to Appendix E.

**Theorem 5.1.** *The set of model updates $\{\Delta\tilde{\boldsymbol{\theta}}_i^e\}_{e=1}^E$, which are uploaded to the server by each client $i \in \{1, \cdots, n\}$ during the training time, as well as their private local model cluster selections satisfy $(\epsilon, \delta)$-DP, where the parameters $\epsilon$ and $\delta$ depend on the DP noise variance $\sigma_{i,DP}^2$ used by the client for DPSGD (Equation $(1)$) and the parameter $\epsilon_{select}$ used for its private cluster selections using exponential mechanism (Equation $(8)$).*

## 6 Evaluation

**Datasets, models and baseline algorithms:** We evaluate our proposed method on three benchamark datasets, including: MNIST (Deng, 2012), FMNIST (Xiao et al., 2017) and CIFAR10 (Krizhevsky, 2009), with heterogeneous data distributions from covariate shift (rotation; $P_i(x)$ varies across clusters) (Kairouz et al., 2021; Werner et al., 2023) and concept shift (label flip; $P_i(y|x)$ varies across clusters) (Werner et al., 2023), *which are the commonly used data splits for clustered FL (see Appendix C)*. We consider four clusters of clients indexed by $m \in \{0, 1, 2, 3\}$ with $\{3, 6, 6, 6\}$ clients, where the smallest cluster is considered as the minority cluster. We compare our method with most recent related DPFL algorithms under an equal total sample-level privacy budget $\epsilon$: 1. Global (Noble et al., 2021): clients run DPSGD locally and send their model updates to the server for aggregation and learning one global model 2. Local (Liu et al., 2022a): clients do not participate FL and learn a local model by running DPSGD on their local data 3. A DP extension of IFCA (Ghosh et al., 2020; Liu et al., 2022a): local loss/accuracy-based clustering performed by clients on existing cluster models 4. MR-MTL (Liu et al., 2022a): uses model personalization to learn one model for each client 5. O-DPCFL: an oracle algorithm which has the knowledge of the true clusters from the *first* round. For R-DPCFL and IFCA, we use exponential mechanism (Rogers & Steinke, 2021), which satisfies zero concentrated DP (z-CDP) (Bun & Steinke, 2016), to privatize clients' local cluster selections. See also Appendix B.3 for details.

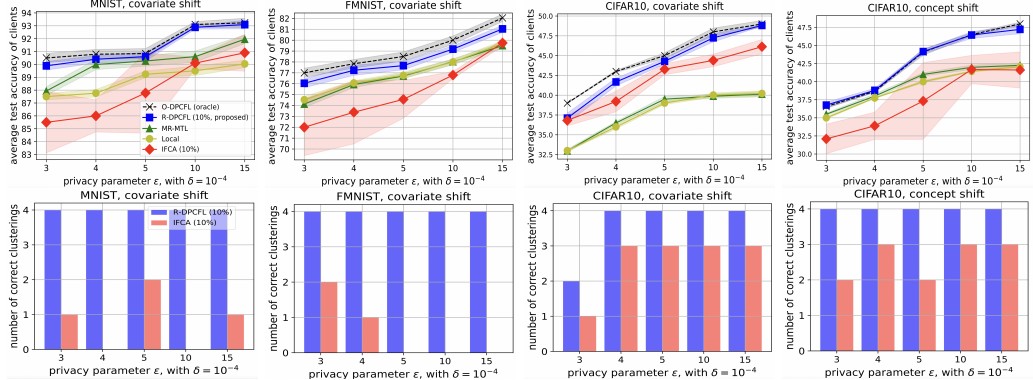

Figure 6: **Top:** Average test accuracy across *all* clients for different total privacy budgets $\epsilon$. Results are from four different runs. 10% means performing local clustering by clients only in 10% of the total number of rounds; i.e., rounds $E_c \leq e \leq E_c + \lfloor \frac{E}{10} \rfloor$ for R-DPCFL and rounds $1 \leq e \leq 1 + \lfloor \frac{E}{10} \rfloor$ for IFCA (see Appendix C.6). **Figure 11 in the appendix includes the** "Global" **baseline** too. **Bottom:** Number of times that R-DPCFL and IFCA successfully detect the underlying cluster structure of all existing clients (out of 4 runs).

## 6.1 Results

Liu et al. (2022a) observed that under sample-level differential privacy (as in this work) and *"mild" data heterogeneity*, federation is more beneficial than local training, because, despite the data heterogeneity across clients, the model aggregation (averaging) on the server diminishes the effect of the DP noise in clients' model updates. However, when there is a *"structured" data heterogeneity* across clients, the level of heterogeneity is remarkable. Hence, learning one global model through FL is not beneficial, as one single model can barely adapt to the high level of data heterogeneity across the clusters. Therefore, in DP clustered FL systems, local training and model personalization can be better options than global training, as they diminish the adverse effect of the high data heterogeneity. Furthermore, *if one can detect the underlying clusters*, one can perform FL in them in parallel and will simultaneously benefit from 1. eliminating the effect of data heterogeneity across clusters; 2. diminishing the effect of DP noise by running FL aggregation on the server within each cluster (as observed by (Liu et al., 2022a)). Hence, *if the clustering task is done correctly*, we can expect a further improvement over local training and model personalization. This is exactly what R-DPCFL aims to do.

**RQ1: How does R-DPCFL perform in practice?** Figure 6 shows the average test accuracy across clients for four datasets. As can be observed, R-DPCFL outperforms the baseline algorithms (thanks to its more efficient clustering method: Figure 6, bottom row). While R-DPCFL performs close to the oracle algorithm, IFCA has a lower performance due to its clustering errors. For instance, IFCA has a clearly low clustering accuracy on MNIST and FMNIST, which leads it to perform even worse than Local and MR-MTL. In contrast, it has a better clustering performance on CIFAR10 (covariate and concept shifts) and outperforms the two baselines. On the other hand, the reason behind the low performance of MR-MTL is that it performs personalization on a global model, which in turn has a low quality due to being obtained from federation across "all" clients (hence adversely affected by the high data heterogeneity). Similarly, Local, which performs close to MR-MTL, cannot outperform R-DPCFL, as it does not benefit from diminishing the effect of DP noise by FL aggregation.

**RQ2: How does the minority cluster benefit from R-DPCFL?** Figure 7 compares different algorithms based on the average test accuracy of the clients belonging to the *minority* cluster. R-DPCFL leads to a better overall performance for the minority clients, by virtue of its correct and robust cluster detection. Correct detection of the minority cluster prevents it from getting mixed with other majority clusters and leads to a utility improvement for its clients. In contrast, IFCA has a lower success rate in detecting the minority cluster (Figure 7, bottom row) and provides a lower overall performance for them. Similarly, Local and MR-MTL lead to a low performance for the minority, as they are conditioned on a global model that is learned from federation across "all" clients and provides a low performance for the minority. Correct detection of minority clusters is important, as failure in detecting them correctly leads to a low performance for the smaller clusters.

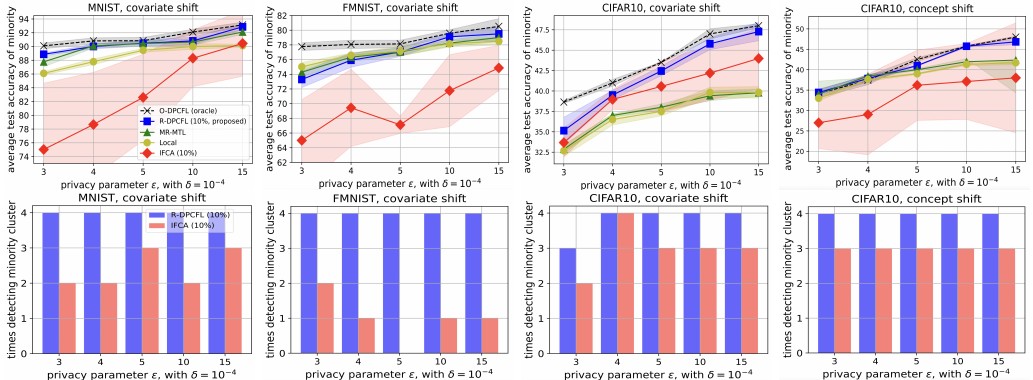

Figure 7: **Top:** Average test accuracy across clients belonging to the *minority* cluster for different total privacy budgets $\epsilon$, and four different runs. **Bottom:** Number of times that R-DPCFL and IFCA successfully detect the minority cluster as a separate cluster (out of 4 runs).

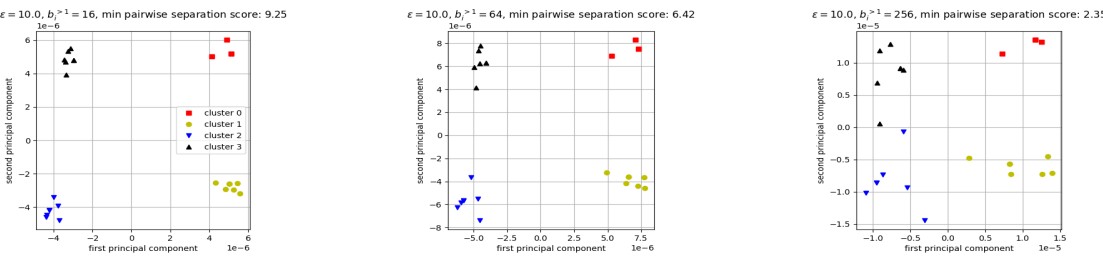

Figure 8: The effect of batch sizes $\{b_i^{>1}\}_{i=1}^n$ on the model updates $\{\Delta\tilde{\boldsymbol{\theta}}_i^1\}_{i=1}^n$ at the end of the first round with full batch sizes in the first round ($\forall i : b_i^1 = N_i$). For a fixed $\epsilon = 10$, the model updates scatter further in space as $b_i^{>1}$ increases and different clusters get less separable. This leads to a decline in the confidence level (MSS score) of the resulting GMM (written on top). Results are on CIFAR10 with covariate shift.

### RQ3: How can we set the hyper-parameters of R-DPCFL?

**Batch size** $b_i^{>1}$: The batch size used by R-DPCFL during rounds $e > 1$ has to be set to a small value, as observed in Figure 3 right. R-DPCFL is not sensitive to this parameter, as long as a small value is chosen for it. For the results reported so far, we used $b_i^{>1} = 32$ for all experiments with R-DPCFL. As we observed in Lemma 4.1 and Figure 3 left, $\texttt{Var}(\Delta\tilde{\boldsymbol{\theta}}_i^1(b_i^1)|\boldsymbol{\theta}^{init})$ is an increasing function of $b_i^{>1}$. More generally, the effect of increasing $b_i^{>1}$ is threefold: 1) increasing noise variance $\texttt{Var}(\Delta\tilde{\boldsymbol{\theta}}_i^1(b_i^1)|\boldsymbol{\theta}^{init})$ (as shown in Figure 3, left) 2) decreasing noise variance $\texttt{Var}(\Delta\tilde{\boldsymbol{\theta}}_i^1(b_i^e)|\boldsymbol{\theta}^{e,0})$ ($e > 1$, as shown in Figure 3, right) 3) decreasing number of gradient steps during each round $e$ for $e > 1$. While the first one is only limited to the first round $e = 1$, the last two affect the remaining $E - 1$ rounds and have conflicting effects on the final accuracy. However, an important point is that finding the true structure of clusters in the first round is a prerequisite for making progress in the next rounds. Therefore, increment in noise variance $\texttt{Var}(\Delta\tilde{\boldsymbol{\theta}}_i^1(b_i^1)|\boldsymbol{\theta}^{init})$ (the first effect) is the most important one. We have demonstrated this effect in Figure 8, which shows that how increasing $b_i^{>1}$ adversely affects the clustering done at the end of the first round. Note how MSS score of the learned GMM increases as $b_i^{>1}$ increases. Therefore, in order to have a reliable client clustering at the end of the first round, we need to keep the value of $b_i^{>1}$ small: the smaller total privacy budget $\epsilon$, the smaller value should be used for $b_i^{>1}$. Following this observation, we have fixed $b_i^{>1}$ to 32 in all our experiments with R-DPCFL.

**The strategy switching time** $E_c$: The strategy switching time $E_c$ can also be set by using the uncertainty metric MPO $\in [0, 1)$. Intuitively, if the learned GMM is not certain about its clustering decisions, R-DPCFL should not rely on its decisions for a large $E_c$, and vice versa. Hence, we can set $E_c$ as a decreasing function of MPO. For instance, $E_c = (1 - \texttt{MPO})\frac{E}{2}$, which is used in this work, means that if a GMM is completely confident about its clusterings, e.g., what happens in Figure 5 right, the server changes the clustering strategy to loss-based

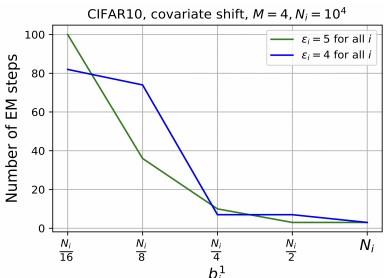 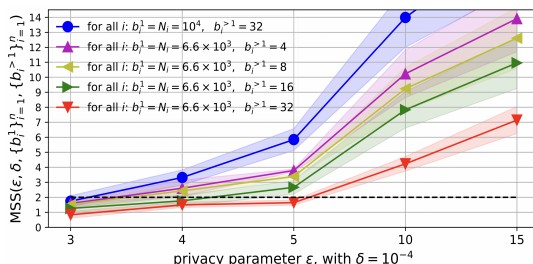

Figure 9: **Left:** The number of `EM` iterations needed for learning the `GMM` decreases as $b_i^1$ increases. Especially, at full batch size ($\forall i : b_i^1 = N_i$), very few iterations of `EM` are needed. The results are obtained on CIFAR10 with covariate shift across $M = 4$ clusters. Using a large enough batch size $b_i^1$ in the first round makes the underlying clusters in $\{\Delta\tilde{\theta}_i^1\}_{i=1}^n$ more distinguishable, and convergence rate of `EM` algorithm for learning the `GMM` increases with $b_i^1$. **Right:** `MSS` score v.s. $\epsilon$ for two different local dataset sizes. A small local dataset size can be compensated for by using smaller batch sizes $\{b_i^{>1}\}_{i=1}^n$ to get a larger `MSS` score.

after the first half of rounds. As the uncertainty increases, this change happens earlier (e.g., when $\epsilon$ is small), and `R-DPCFL` slowly gets close to the existing loss-based clustering methods like `IFCA` (Ghosh et al., 2020).

**Number of clusters $M$:** Finally, we can use `MSS` score of the learned `GMM` to set the number of underlying clusters $M$ when it is unknown at the beginning. More specifically, we use the value of $M$ for which the uncertainty of the learned `GMM` is minimized. We refer to Appendix C.9 for further details. We remind that the `DP` extension of `IFCA` needs to know the true number of clusters $M$ beforehand.

**RQ4: How does $b_i^1$ affect the convergence rate of `EM` for learning the `GMM` at round $e = 1$?** Note that the only computational overhead of `R-DPCFL` compared to the `DP` extension of `IFCA` is that it learns a `GMM` at the end of the very first round. Importantly, as we proved in Theorem 4.3 this overhead drops very quickly by simulating large batch sizes in the very first round (by using the gradient accumulation technique with zero overheads). The results in Figure 9 (left) supports Theorem 4.3 by showing that if $b_i^1$ is large enough, the convergence rate of `EM` algorithm for learning the `GMM` at the end of the first round increases with $b_i^1$. Hence, using large batch sizes in the first round reduces the computational complexity of learning the `GMM`.

**RQ5: What if clients have small local datasets?** While we envision the proposed approach being more applicable to cross-silo `FL`, where clients' datasets are large, it is still worth exploring how beneficial it can be under scarce local data. As learned in the previous sections, the `MSS` score of the learned `GMM` can strongly predict whether the underlying true clusters will be detected: an `MSS` above 2 almost always yields to correct detection of the underlying clusters (see Figure 13 for experimental results). For a fixed $(\epsilon, \delta)$, larger $\{b_i^1\}_{i=1}^n$ and smaller $\{b_i^{>1}\}_{i=1}^n$ increase `MSS` (Lemma 4.1). When $b_i^1 = N_i$ (i.e. full batch sizes for all $i$ in the first round), smaller local datasets result in lower confidence in the learned `GMM`. Nevertheless, this can be compensated for by using even smaller $\{b_i^{>1}\}_{i=1}^n$. Figure 9 (right) compares two different dataset sizes under varying $\epsilon$. As observed, reducing $\{b_i^{>1}\}_{i=1}^n$ can compensate for smaller local dataset sizes by leading to less noisy model updates $\{\Delta\tilde{\theta}_i^1\}_{i=1}^n$. This improves the `MSS` score of the learned `GMM` and the clustering accuracy. Finally, we refer to Appendix H for some hints about using data augmentation in the first round.

# 7 Conclusion

We proposed a sample-level `DP` clustered `FL` algorithm addressing systems with structured data heterogeneity. By clustering clients based on both their model updates and training loss/accuracy values, and using large initial batch sizes, our approach enhances clustering accuracy and mitigates the disparate impact of `DP` on utility, all with minimal computational overhead. The proposed approach is robust to `DP` noise and has easy parameter selection. While envisioned for `DPFL` systems with large clients' datasets, the method is capable of compensating for moderate dataset sizes by using smaller batch sizes after the first round. A future research direction is addressing this limitation and extending the approach to be suitable for scarce data scenarios.

## 8 Acknowledgment

Funding support for project activities has been partially provided by Canada CIFAR AI Chair, Facebook award, Google award, FRQNT scholarship and MEI award. We also express our gratitude to Compute Canada for their support in providing facilities for our evaluations.

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
