# Appendix for *Differentially Private Clustered Federated Learning*

## A    Notations

Table 1 summarizes the notations used in the paper.

Table 1: Used notations.

| | |
|---|---|
| $n$ | number of clients, which are indexed by $i$ |
| $x_{ij}, y_{ij}$ | $j$-th data point of client $i$ and its label |
| $\mathcal{D}_i, N_i$ | local train set of client $i$ and its size |
| $\mathcal{D}_{i,aug}$ | augmented local train set of client $i$ |
| $\mathcal{B}_i^{e,t}$ | the train data batch used by client $i$ in round $e$ and at the $t$-th gradient update |
| $b_i^e$ | batch size of client $i$ in round $e$: $|\mathcal{B}_i^{e,t}| = b_i^e$ |
| $b_i^1$ | batch size of client $i$ during the first round $e = 1$ |
| $b_i^{>1}$ | set of batch sizes of client $i$ during the rounds $e > 1$ |
| $\epsilon, \delta$ | desired DP privacy parameters |
| $E$ | total number of communication rounds in the DPFL system, indexed by $e$ |
| $\boldsymbol{\theta}_m^e$ | model parameter for cluster $m$, at the beginning of global round $e$ |
| $K$ | number of local train epochs performed by clients during each global round $e$ |
| $\eta_l$ | the common learning rate used for DPSGD |
| $h$ | predictor function, e.g., CNN model, with parameter $\boldsymbol{\theta}$ |
| $\ell$ | cross entropy loss |
| $s(i)$ | the true cluster of client $i$ |
| $R^e(i)$ | the cluster assigned to client $i$ in round $e$ |
| $\boldsymbol{\theta}_i^{e,0}$ | the starting model parameter of client $i$ at the beginning of round $e$ |
| $\Delta \tilde{\boldsymbol{\theta}}_i^e(b_i^e)$ | the noisy update of client $i$ at round $e$, starting from $\boldsymbol{\theta}_i^{e,0}$, with batch size $b_i^e$ |
| $(\sigma_i^e(b_i^e))^2$ | conditional variance of the update $\Delta \tilde{\boldsymbol{\theta}}_i^e$ of client $i$: $\text{Var}(\Delta \tilde{\boldsymbol{\theta}}_i^e(b_i^e)|\boldsymbol{\theta}_i^{e,0})$ |
| $\mu_m^*(b^1)$ | the center of the $m$-th cluster (when all use batch size $b^1$ in the first round) |
| $\Sigma_m^*(b^1)$ | the covariance matrix of the $m$-th cluster (when all use batch size $b^1$ in the first round) |
| $\alpha_m^*$ | the prior probability of the $m$-th cluster |

## B    Background

### B.1    Rényi differential privacy (RDP)

We have used a relaxation of Differential Privacy, named Rényi DP (RDP) for tight privacy accounting of different algorithms (Mironov, 2017). It is defined as follows:

**Definition B.1** (Rényi Differential Privacy (RDP) (Mironov, 2017))**.** *A randomized mechanism $\mathcal{M} : \mathcal{A} \to \mathcal{R}$ with domain $\mathcal{D}$ and range $\mathcal{R}$ satisfies $(\alpha, \epsilon)$-RDP with order $\alpha$ if for any two adjacent inputs $\mathcal{D}, \mathcal{D}' \in \mathcal{A}$, which differ only by a single record (by replacement),*

$$D_\alpha\big(\mathcal{M}(\mathcal{D})||\mathcal{M}(\mathcal{D}')\big) \le \epsilon,$$

where $D_\alpha(P||Q)$ is the Rényi divergence between distributions $P$ and $Q$:

$$D_\alpha(P||Q) := \frac{1}{\alpha - 1} \log \mathbb{E}_{x \sim p}\left[\left(\frac{P(x)}{Q(x)}\right)^{\alpha-1}\right]. \tag{6}$$

For $\alpha = 1$, we have $D_1(P||Q) := \mathbb{E}_{x \sim p}\left[\log\left(\frac{P(x)}{Q(x)}\right)\right]$, which is the KL divergence between $P$ and $Q$. RDP is conveniently linearly composable, as explained in the following theorem.

**Theorem B.2** (Linear Composition of RDP (Mironov, 2017)). *If mechanism $\mathcal{M}_i$ satisfies $(\alpha, \epsilon_i)-$RDP for $i = 1, \ldots, k$, then the composed mechanism $\mathcal{M}_1 \circ \ldots \circ \mathcal{M}_k$ satisfies $(\alpha, \sum_{i=1}^{k} \epsilon_i)-$RDP.*

Therefore, if an algorithm has $E$ steps (e.g., $E$ batch gradient update steps) and each satisfies $(\alpha, \epsilon)$-RDP, the algorithm will satisfy $(\alpha, E\epsilon)$-RDP. RDP can also be used for composition of *heterogeneous* private mechanisms, e.g., for accounting privacy of R-DPCFL, which uses different batch sizes in the first and the subsequent rounds. The following lemma is about conversion of $(\alpha, \epsilon)$-RDP to standard $(\epsilon, \delta)$-DP (Definition 3.1).

**Lemma B.3** (Converting RDP (Bun & Steinke, 2016; Canonne et al., 2020)). *If a mechanism $\mathcal{M}$ satisifes $(\alpha, \epsilon(\alpha))$-RDP, then for any $\delta > 0$, it satisfies $(\epsilon(\delta), \delta)$-DP, where*

$$\epsilon(\delta) = \inf_{\alpha > 1} \epsilon(\alpha) + \frac{1}{\alpha - 1} \log\left(\frac{1}{\alpha\delta}\right) + \log\left(1 - \frac{1}{\alpha}\right). \tag{7}$$

The Gaussian mechanism satisfies $(\alpha, \epsilon)$-RDP, based on the following Proposition from (Mironov, 2017):

**Proposition B.4.** *If $f : \mathcal{A} \to \mathcal{R}$ has sensitivity c, then its randomization with a Gaussian mechanism with noise variance $\sigma_{DP}^2$ satisfies $(\alpha, \frac{\alpha c^2}{2\sigma_{DP}^2})$-RDP.*

Some accounting routines have been implemented in open source libraries for accounting privacy of RDP mechanisms. We use TensorFlow Privacy implementation (McMahan et al., 2019) in this work.

## B.2 Zero concentrated differential privacy (z-CDP)

Another relaxed definition of differential privacy is zero concentrated differential privacy (z-CDP) (Bun & Steinke, 2016). Being $\rho$ z-CDP is equivalent to being $(\alpha, \rho\alpha)$-RDP simultaneously for all $\alpha > 1$. Therefore, standard RDP accountants, e.g., the aforementioned TensorFlow Privacy RDP accountant (McMahan et al., 2019), can be use for accounting mechanism satisfying z-CDP as well.

## B.3 Exponential mechanism for private selection

Exponential Mechanism is a standard for private selection from a set of candidates. The selection is based on a score, which is assigned to every candidate (Rogers & Steinke, 2021). Let us assume there is a private dataset $\mathcal{D}$ and a score function $s : \mathcal{D} \times [M] \to \mathbb{R}$, which evaluates a set of $M$ candidates on the dataset $\mathcal{D}$. The goal is to select the candidate with the highest score, i.e., $\arg\max_{m \in [M]} s(\mathcal{D}, m)$. Exponential mechanism performs this selection privately as follows. It sets the probability of choosing any candidate $m \in [M]$ as:

$$\Pr[m] = \frac{\exp(\frac{\epsilon_{\text{select}}}{2\Delta} \cdot s(\mathcal{D}, m))}{\sum_{m' \in [M]} \exp(\frac{\epsilon_{\text{select}}}{2\Delta} \cdot s(\mathcal{D}, m'))}, \tag{8}$$

where $\Delta$ is the sensitivity of the scoring function $s$ to the replacement of a data sample in $\mathcal{D}$. It can be shown that the private selection performed by exponential mechanism satisfies $\frac{1}{8}\epsilon_{\text{select}}^2$ z-CDP with respect to $\mathcal{D}$ (Bun & Steinke, 2016), which from the last paragraph, we know satisfies $(\alpha, \frac{\alpha}{8}\epsilon_{\text{select}}^2)$-RDP for $\alpha > 1$. We implement exponential mechanism by noisy selection with Gumbel noise: we add independent noises from Gumbel distribution with scale $\frac{2\Delta}{\epsilon_{\text{select}}}$ to candidate scores $s(\mathcal{D}, m)$, for $m \in [M]$, and select the candiate with the maximum noisy score. The larger the sensitivity $\Delta$ of score $s$ to replacement of a single sample in $\mathcal{D}$, the required larger noise scale. For further details about how we implement exponential mechanism for IFCA and R-DPCFL, see Appendix C.6.

## B.4 Privacy budgeting

In order to have a fair comparison between our algorithm and the baselines, we align them all to have the same "total" privacy budget $\epsilon$ and satisfy $(\epsilon, \delta)$-DP for a fixed $\delta$. In order to account the privacy of an algorithm, we compose the RDP guarantees of all private operations in the algorithm and then convert the resulting RDP

guarantee to approximate $(\epsilon, \delta)$-DP using Lemma B.3. The `DPSGD` performed by different algorithms for local training benefits from privacy amplification by subsampling (Mironov et al., 2019). Algorithms that have privacy overheads, e.g., `IFCA` and `R-DPCFL` which need to privatize their local clustering as well, will have less privacy budget left for training. In other words, for the same total privacy budget $\epsilon$, `IFCA` and `R-DPCFL` will use a larger amount of noise when running `DPSGD`, compared to `MR-MTL` that has zero privacy overhead.

## C  Experimental setup

### C.1  Datasets

**Data split:**  We use three datasets MNIST, FMNIST and CIFAR10, and consider a distributed setting with 21 clients. In order to create majority and minority clusters, we consider 4 clusters with different number of clients $\{3, 6, 6, 6\}$ (21 clients in total). The first cluster with the minimum number of clients is the "minority" cluster, and the last three are the "majority" ones. The data distribution $P(x, y)$ varies across clusters. We use two methods for making such data heterogeneity: 1. **covariate shift** 2. **concept shift**. In covariate shift, we assume that features marginal distribution $P(x)$ differs from one cluster to another cluster. In order to create this variation, we first allocate samples to all clients in an *uniform* way. Then we rotate the data points (images) belonging to the clients in cluster $k$ by $k * 90$ degrees. For concept shift, we assume that conditional distribution $P(y|x)$ differs from one cluster to another cluster, and we first allocate data samples to clients in a uniform way, and flip the labels of the points allocated to clients: we flip $y_{ij}$ (label of the $j$-th data point of client $i$, which belongs to cluster $k$) to $(y_{ij} + k) \ mod \ 10$, The local datasets are balanced–all users have the same amount of training samples. The local data is split into train and test sets with ratios 80%, and 20%, respectively. In the reported experimental results, all users participate in each communication round.

Table 2: CNN model for classification on MNIST/FMNIST datasets

| Layer | Output Shape | # of Trainable Parameters | Activation | Hyper-parameters |
|-------|--------------|---------------------------|------------|------------------|
| Input | $(1, 28, 28)$ | 0 | | |
| Conv2d | $(16, 28, 28)$ | 416 | ReLU | kernel size =5; strides=$(1, 1)$ |
| MaxPool2d | $(16, 14, 14)$ | 0 | | pool size=$(2, 2)$ |
| Conv2d | $(32, 14, 14)$ | 12,832 | ReLU | kernel size =5; strides=$(1, 1)$ |
| MaxPool2d | $(32, 7, 7)$ | 0 | | pool size=$(2, 2)$ |
| Flatten | 1568 | 0 | | |
| Dense | 10 | 15,690 | ReLU | |
| Total | | 28,938 | | |

### C.2  Models and optimization

We use a simple 2-layer CNN model with ReLU activation, the detail of which can be found in Table 2 for MNIST and FMNIST. Also, we use the residual neural network (ResNet-18) defined in (He et al., 2015), which is a large model. To update the local models allocated to each client during each round, we apply `DPSGD` (Abadi et al., 2016) with a noise scale $z$ which depends on some parameters, as in Lemma 4.1.

Table 3: Details of the used datasets in the main body of the paper. ResNet-18 is the residual neural networks defined in He et al. (2015). CNN: Convolutional Neural Network defined in Table 2.

| Datasets | Train set size | Test set size | Data Partition method | # of clients | Model | # of parameters |
|----------|----------------|---------------|-----------------------|--------------|-------|-----------------|
| MNIST | 48000 | 12000 | covariate shift | $\{3, 6, 6, 6\}$ | CNN | 28,938 |
| FMNIST | 48000 | 12000 | covariate shift | $\{3, 6, 6, 6\}$ | CNN | 28,938 |
| CIFAR10 | 40000 | 10000 | covariate and concept shift | $\{3, 6, 6, 6\}$ | ResNet-18 | 11,181,642 |

In order to simulate a `FL` setting, where clients (silos) have large local datasets and there is a structured data heterogeneity across clusters, we split the full dataset between the clients belonging to each cluster. This way,

each client gets $8,000$ train and $1,666$ test samples for MNIST and FMNIST. Also, each client gets $10,000$ and $1,666$ train and test samples for CIFAR10 dataset (both covariate shift and concept shift).

## C.3 Baseline selection

When extending existing model personalization and clustered FL algorithms to DPFL settings, we are mostly interested in those with little to no additional local dataset queries to prevent extra noise for DPSGD under a fixed total privacy budget $\epsilon$. For instance, the family of mean-regularized multi-task learning methods (MR-MTL) (Evgeniou & Pontil, 2004; Hanzely et al., 2020; Hanzely & Richtárik, 2021; Dinh et al., 2022) provide model personalization *without an additional privacy overhead*. Despite this, it is noteworthy that MR-MTL relies on optimal hyperparameter tuning which leads to a potential privacy overhead (Liu et al., 2022a; Liu & Talwar, 2018; Papernot & Steinke, 2022). While resembling MR-MTL, Ditto (Li et al., 2021) has extra local computations, which makes it a less attractive personalization algorithm. Hence, we adopt MR-MTL (Liu et al., 2022a) as a baseline personalization algorithm. Similarly, multi-task learning algorithms of Smith et al. (2017) and Marfoq et al. (2021) as well as gradient-based clustered FL algorithm of Sattler et al. (2019) benefit from additional training and training restarts, which lead to high privacy overhead for them, making them less attractive. In contrast, the aforementioned loss-based clustered FL algorithms (Mansour et al., 2020; Ghosh et al., 2020; Ruan & Joe-Wong, 2021) can be managed to have a low privacy overhead (see Appendix C.6), and we use it as a clustered DPFL baseline.

## C.4 MR-MTL formulation

The objective function of Mean-Regularzied Multi-Task Learning (MR-MTL) can be expressed as:

$$\min_{\boldsymbol{\theta}_i, i \in \{1, \cdots, n\}} \sum_{i=1}^{n} g_i(\boldsymbol{\theta}_i) \quad \text{with} \quad g_i(\theta_i) = f_i(\boldsymbol{\theta}_i) + \frac{\lambda}{2} \|\boldsymbol{\theta}_i - \bar{\boldsymbol{\theta}}\|_2^2, \tag{9}$$

where $\bar{\boldsymbol{\theta}} = \frac{1}{n} \sum_{i=1}^{n} \boldsymbol{\theta}_i$ is the average model parameter across clients and $f_i(\boldsymbol{\theta}_i)$ is the loss function of personalized model parameter $\theta_i$ of client $i$ on its local dataset $\mathcal{D}_i$. With $\lambda = 0$, MR-MTL reduces to local training. A larger regularization term $\lambda$ encourages local models to be closer to each other. However, MR-MTL may not recover FedAvg (McMahan et al., 2017) as $\lambda \to \infty$. See algorithm A1 in (Liu et al., 2022a) for more details about MR-MTL.

## C.5 Tuning hyperparameters of baseline algorithms

Appendix C.3 explains our criteria for baseline selection. We compare our R-DPCFL algorithm, which benefits from robust clustering, with four baseline algorithms, including: 1) DPFedAvg (Noble et al., 2021), which learns one global model for all clients, and is called Global in the paper 2) Local, in which clients do not participate FL and run DPSGD locally to train a model solely on their local dataset 3) MR-MTL personalized FL algorithm (Liu et al., 2022a), which learns a global model and one personalized model for each client 4) A DP extension of the clustered FL algorithm IFCA (Ghosh et al., 2020) to DPFL systems enhanced with exponential mechanism (see Appendix C.6) 5) An oracle algorithm, which has the knowledge of the true underlying clients' clusters, which we call O-DPCFL.

For all algorithms and all datasets, we set total number of rounds $E$ to 200 and per-round number of local epochs $K$ to 1. Following (Abadi et al., 2016), we set the batch size of each client such that the number of batches per epoch is in the same order as the total number of epochs: $N_i/b_i^e = E \cdot K = 200$. For MNIST and FMNIST, this leads to batch sizes $b_i^e = 32$ for all clients $i$ and every round $e$ for the baseline algorithms. For CIFAR10 (covariate shift and concept shift), this leads to batch size $b_i^e = 64$ for all clients $i$ and every round $e$ for the baseline algorithms. As explained in Section 4.2.1, While R-DPCFL uses full batch sizes in the first round (i.e., $b_i^1 = N_i$ for all $i$), it needs to use small batch sizes in the next rounds (small $b_i^{>1}$).

Having determined the batch size for all algorithms, clipping threshold $c$ and learning rate $\eta_l$ are determined via a grid search on clients' validation sets. For each algorithm and each dataset, we find the best learning rate

from a grid: the one which results in the highest average accuracy at the end of FL training on a validation set with size 1666 samples for each client. We use the grid $\eta_l \in$ {5e-4, 1e-3, 2e-3, 5e-3, 1e-2, 2e-2, 5e-2, 1e-1} for all datasets and all algorithms. Similarly, we use the grid $c \in$ {1, 2, 3, 4, 5} for setting the clipping threshold for all datasets and all algorithms based on the clients' validation sets.

### C.6 Implementation of private local clustering for IFCA and R-DPCFL

In every round of IFCA and during the rounds $e > E_c$ of R-DPCFL, the server sends $M$ cluster models to all clients, and they evaluate them on their local datasets. Then, each client $i$ selects the model with the lowest loss on its local dataset $\mathcal{D}_i$, trains it for $K$ local epochs and sends the result back to the server. This model selection performed by each client can lead to privacy leakage w.r.t its local dataset, if it is not privatized. In order to protect data privacy, clients need to privatize their local cluster selection by using exponential mechanism and accounting its privacy using z-CDP, explained in Appendix B.3. Assuming a total privacy budget $\epsilon$ for a client $i$, it has to split the budget between private clustering and DPSGD. Naive split of privacy budget can lead to very noisy DPSGD steps or very noisy local cluster selection by clients. Following Liu et al. (2022a), we use two strategies to mitigate the privacy overhead of local clustering performed by IFCA and R-DPCFL:

- **Clients use models' train accuracy, instead of loss, as the score function for model selection:** clients use train accuracy as score function $s(\mathcal{D}_i, m)$ evaluating cluster model $m$ on client $i$'s dataset. The reason is that while, loss function has practically an unbounded sensitivity to individual samples in the clients' datasets, model accuracy is a low-sensitivity function, espcially in cross-silo FL settings with large local datasets. More specifically, let us assume client $i$ with local dataset $\mathcal{D}_i$ (which has size $N_i$) uses models' accuracy on $\mathcal{D}_i$ for model selection. It can be shown that under all add/remove/replace notions of dataset neighborhood, sensitivity of model accuracy (as score function) is bounded as follows (Liu et al., 2022a):

$$\Delta_{acc} = \max_{m \in [M]} \max_{\mathcal{D}_i, \mathcal{D}_i'} |s(\mathcal{D}_i, m) - s(\mathcal{D}_i, m)| \leq \frac{1}{N_i - 1}. \tag{10}$$

Since local dataset sizes are usually large, especially in cross-silo FL, the sensitivity of model accuracy is much smaller than that of model loss. Therefore, following Liu et al. (2022a), we set the per-round privacy budget of private model selection to a very smalle value $\epsilon_{\text{select}} = 0.03 \cdot \epsilon$ (3% of the total privacy budget). Yet, the cost of private selection by clients can grow quickly if clients naively run local clustering for "many" rounds. Therefore, we use the following strategy as well. It is noteworthy that in our experiments, we observed that IFCA baseline algorithm performs better when clients use model train accuracy (instead of train loss) for cluster selection.

- **Reduce the number of rounds with local clustering on clients' side:** Clients run local clustering for less rounds. Following Liu et al. (2022a), we let clients run local clustering for the 10% of the total number of rounds $E$. For example, IFCA runs local clustering during the first $\lfloor \frac{E}{10} \rfloor$ rounds, and fixes clients' cluster assignments afterwards. Similarly, R-DPCFL lets clients run local clustering during rounds $E_c \leq e \leq E_c + \lfloor \frac{E}{10} \rfloor$, and fixes clients' cluster assignments afterwards.

The privacy overhead of private model selection can still grow and leave a low privacy budget for training with DPSGD. Choosing a small selection budget $\epsilon_{\text{select}}$ leaves most of the total privacy budget $\epsilon$ for training with DPSGD, but leads to noisy and inaccurate cluster selection by clients. Similarly, a large $\epsilon_{\text{select}}$ leads to more noisy gradient steps by DPSGD.

### C.7 DP privacy parameters

For each dataset, 5 different values of $\epsilon$ (the total privacy budget) from set $\{3, 4, 5, 10, 15\}$ are used. We fix $\delta$ for all experiments to $10^{-4}$, which satisfies $\delta < N_i^{-1}$ for every client $i$. We use the Rényi DP (RDP) privacy accountant (TensorFlow privacy implementation (McMahan et al., 2019)) during the training time. This

accountant is able to handle the difference in the batch size of R-DPCFL between the first round $e = 1$ and the next rounds $e > 1$ by accounting the composition of the corresponding *heterogeneous* private mechanisms.

## C.8   Gaussian mixture model

We use the Gaussian mixture model of Scikitlearn, which can be found here: `https://scikit-learn.org/dev/modules/generated/sklearn.mixture.GaussianMixture.html`. The GMM model has three hyper-parameters:

1) parameter initialization, which we set to "`k-means++`". This is because this type of initialization leads to both low time to initialize and low number of EM iterations for the GMM to converge (Arthur & Vassilvitskii, 2007; Biernacki et al., 2003).

2) Type of the covariance matrix, which we set to "`spherical`", i.e., each component has a diagonal covariance matrix with a single value as its diagonal elements. This is in accordance with Equation (24) and that we know the covariance matrices should be diagonal.

3) Finally, the number of components (clusters) is either known or it is unknown. In the latter case, we have explained in Appendix C.9 how we can find the true number of clusters by using the confidence level (MSS) of the GMM model learned at the end of the first round.

## C.9   Finding the number of clusters $M$ when it is unknown

Knowing the number of clusters is broadly accepted and applied in the clustered FL literature (Ghosh et al., 2020; Ruan & Joe-Wong, 2021; Briggs et al., 2020). The used baseline algorithms have also made this assumption. Yet, techniques to determine the number of clusters can enable our approach to be more widely adopted. In this section, we show that how we can find the true number of clusters ($M$) when it is not given. Our method relies on the MSS score (confidence level): $\text{MSS} = \min_{m,m'} \hat{\text{SS}}(m, m') \in [0, +\infty)$. Consider the Figure 5 right as an example. There is a good separation between the $M = 4$ existing clusters, thanks to clients using full batch sizes in the first round. Fitting a GMM with 4 components to the model updates results in the highest MSS for the learned GMM model: remember that MSS was the maximum pairwise separation score between the different components of the learned GMM. In contrast, if we fit a GMM with 3 components (less than the true number of components) to the same model updates in the figure, then two clusters will be merged into one component (for examples clusters 0 and 1) leading to a high radius for one of the three components of the resulting GMM. This leads to a low MSS (confidence level) for the resulting GMM. Similarly, if we fit a GMM with 5 components, one of the four clusters (for example cluster 1) will be split between two of the 5 components (call them $m$ and $m'$), which leads to a low inter-component distance ($\Delta_{m,m'}$) for the pair of components. This also leads to a low MSS for the resulting GMM. However, **fitting a GMM with $M = 4$ components leads to a well separation between all the true components and maximizes the resulting MSS**. Based on this very intuitive observation, we propose the following method for setting $m$ at the end of the first round: We select the number of clusters/components, which leads to the maximum MSS for the resulting GMM. More specifically:

$$M = \arg\max_{m \in S} \text{MSS}\Big(\text{GMM}(\Delta\tilde{\boldsymbol{\theta}}_1^1, \ldots, \Delta\tilde{\boldsymbol{\theta}}_n^1; m)\Big), \text{(line 10 of Algorithm 1)} \tag{11}$$

where $S$ is a set of candidate values for $M$: at the end of the first round and on the server, we learn one GMM for each candidate value in $S$ on the same received model updates $\{\Delta\tilde{\boldsymbol{\theta}}_i^1\}_{i=1}^n$. Finally, we choose the value resulting in the GMM with the highest MSS (confidence). Therefore, this method is run on the server and does not incur any additional privacy overheads. It is also noteworthy that we know from Lemma 4.2 that learning the GMM does not incur much computational cost when large enough (and small enough) batch sizes are used in the first round (subsequent rounds).

We have evaluated this method on multiple data splits and different privacy budgets ($\epsilon$) on CIFAR10, MNIST and FMNIST. The method could predict the number of underlying clusters with 100% accuracy for the MNIST and FMNIST datasets for all values of $\epsilon$. Results for CIFAR10 are shown in Figure 10. As can be

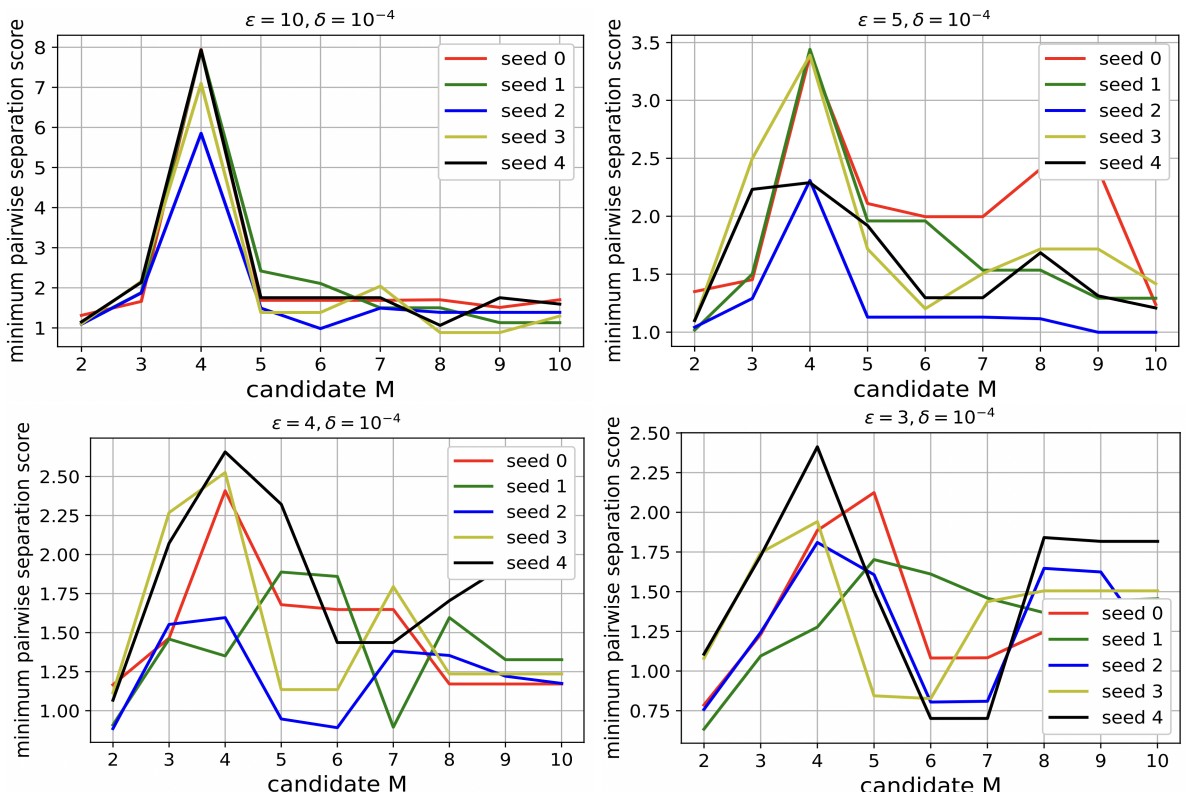

Figure 10: The minimum pairwise separation score (MSS) or confidence of the GMM learned on $\{\Delta\tilde{\boldsymbol{\theta}}_i^1\}_{i=1}^n$ peaks at the true cluster number, which is equal to 4 in all the plots above. Each figure is for a different value of $\epsilon$ (mentioned on top of each figure), and are obtained on CIFAR10 with covariate shift (rotation) across clusters, and 5 different random data splits (5 seeds). All the results are obtained with full batch sizes in the first round and $b_i^{>1} = 32$ for all $i$. We can use this observation as a method to find the true number of clusters ($M$) when it is not given. For larger $\epsilon$, this method work perfectly and even when $\epsilon$ is too small, e.g., $\epsilon = 3$, this method works well and predicts the true number of clusters correctly most of the times: 3 out of the 5 curves in the bottom right plot have a peak at $M = 4$ (the true cluster number). and the other 2 curves predict 5 as the true number, which is the closest and the best alternative for the true value $M = 4$.

observed, the method has made only one mistake for $\epsilon = 4$ (seed 1) and two mistakes for $\epsilon = 3$ (seeds 0 and 1), out of 20 total experiments. Even in those three cases, it has predicted $M$ as 5, which is closest to the true value ($M = 4$) and does not lead to much performance drop (because having $M = 5$ splits an existing cluster into two and it is better than predicting for example $M = 3$, which results in "mixing" two clusters with heterogeneous data). Even in this cases, we can improve the prediction accuracy further by using smaller values of $b_i^{>1}$ (simultaneously with full batch sizes $b_i^1 = N_i$), e.g., $b_i^{>1} = 16$ or $b_i^{>1} = 8$, instead of $b_i^{>1} = 32$ in the figure above. This improvement happens as reducing $b_i^{>1}$ constantly enhances the separation between the underlying components (See Figure 8), which leads to higher accuracy in prediction of the true $M$.

Finally, note that none of the existing baseline algorithms has such an easy and applicable strategy for finding $M$. This shows another useful feature of the proposed R-DPCFL, which makes it more applicable to DP clustered FL settings.

## D   More experimental results

The results shown in Figure 11 and Figure 12 include the results for the Global baseline and are the more complete versions of the figures in the paper (Figure 6 and Figure 7).

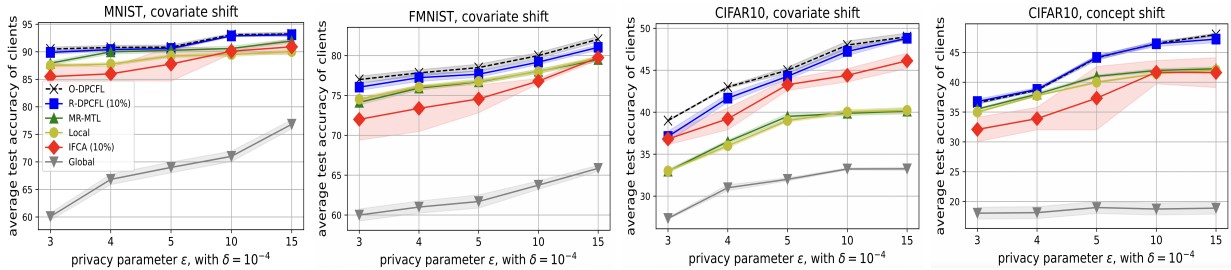

Figure 11: Average test accuracy across clients for different total privacy budgets $\epsilon$ (results are obtained from 4 different random seeds). 10% means performing loss-based clustering by clients only in 10% of the total rounds ($E$).

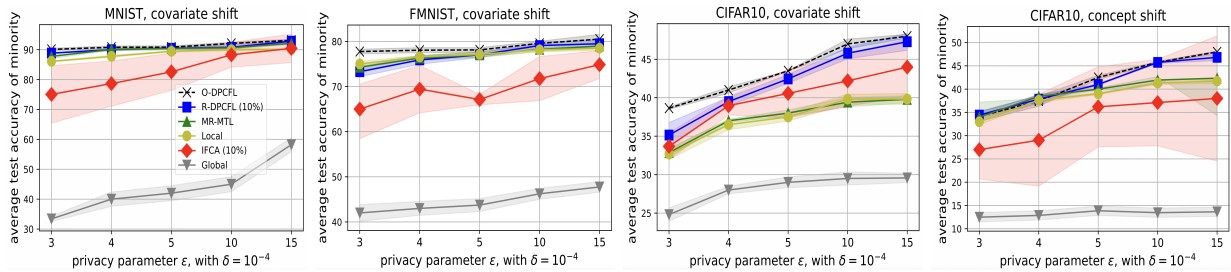

Figure 12: Average test accuracy across clients belonging to the minority cluster for different total privacy budgets $\epsilon$ (results are obtained from 4 different random seeds). 10% means performing loss-based clustering by clients only in 10% of the total rounds ($E$).

Figure 13 shows how the `MSS` score of the learned `GMM` at the first round can be indicative of whether the true clients' clusters will be detected correctly or not. An `MSS` score above 2 almost always yields to correct detection of all clusters.

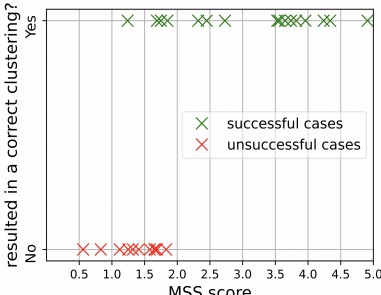

Figure 13: The `MSS` score of the learned `GMM` is indicative of whether the true underlying clusters will be detected or not: an `MSS` score above 2 always leads to correct detection of clusters. Each point is the result of one independent experiment.

# E   Proofs

## E.1   Proof of Lemma 4.1

**Lemma 4.1.** *Let us assume $\boldsymbol{\theta}_i^{e,0}$ is the starting model parameter for client $i$ at the beginning of round $e$. After $K$ local epochs with step size $\eta_l$, the client generates the noisy `DP` model update $\Delta\tilde{\boldsymbol{\theta}}_i^e(b_i^e)$ at the end of the round. The amount of noise in the resulting model update can be quantified as:*

$$(\sigma_i^e(b_i^e))^2 := \mathtt{Var}(\Delta\tilde{\boldsymbol{\theta}}_i^e(b_i^e)|\boldsymbol{\theta}_i^{e,0}) \approx K \cdot N_i \cdot \eta_l^2 \cdot \frac{pc^2 z_i^2(\epsilon, \delta, b_i^1, b_i^{>1}, N_i, K, E)}{(b_i^e)^3}. \tag{2}$$

*Proof.* The following proof has some common parts with similar results in (Malekmohammadi et al., 2024). We consider two illustrative scenarios:

**Scenario 1: the clipping threshold $c$ is effective for all samples in a batch:** in this case we have: $\forall j \in \mathcal{B}_i^{e,t} : c < \|g_{ij}(\boldsymbol{\theta})\|$. Also, we know that the two sources of randomness (i.e., stochastic and Gaussian noise) are independent, thus their variances can be summed up. Let us assume that $E[\bar{g}_{ij}(\boldsymbol{\theta})] = G_i(\boldsymbol{\theta})$ for all samples $j$. From Equation (1), we can find the mean of each *batch gradient* $\tilde{g}_i^{e,t}(\boldsymbol{\theta})$ (of client $i$ in round $e$ and gradient step $t$) as follows:

$$\mathbb{E}[\tilde{g}_i^{e,t}(\boldsymbol{\theta})] = \frac{1}{b_i^e}\sum_{j\in\mathcal{B}_i^{e,t}}\mathbb{E}[\bar{g}_{ij}(\boldsymbol{\theta})] = \frac{1}{b_i^e}\sum_{j\in\mathcal{B}_i^{e,t}}G_i(\boldsymbol{\theta}) = G_i(\boldsymbol{\theta}). \tag{12}$$

Also, from Equation (1), we can find the variance of each *batch gradient* $\tilde{g}_i^{e,t}(\boldsymbol{\theta})$ (of client $i$ in round $e$ and gradient step $t$) as follows:

$$
\begin{aligned}
\sigma_{i,\tilde{g}}^2(b_i^e) := \mathtt{Var}[\tilde{g}_i^{e,t}(\boldsymbol{\theta})] &= \mathtt{Var}\left[\frac{1}{b_i^e}\sum_{j\in\mathcal{B}_i^{e,t}}\bar{g}_{ij}(\boldsymbol{\theta})\right] + \frac{p\sigma_{i,\mathrm{DP}}^2}{(b_i^e)^2} \\
&= \frac{1}{(b_i^e)^2}\left(\mathbb{E}\left[\left\|\sum_{j\in\mathcal{B}_i^{e,t}}\bar{g}_{ij}(\boldsymbol{\theta})\right\|^2\right] - \left\|\mathbb{E}\left[\sum_{j\in\mathcal{B}_i^{e,t}}\bar{g}_{ij}(\boldsymbol{\theta})\right]\right\|^2\right) + \frac{pc^2 z_i^2(\epsilon_i, \delta_i, b_i^1, b_i^{>1}, N_i, K, E)}{(b_i^e)^2} \\
&= \frac{1}{(b_i^e)^2}\left(\mathbb{E}\left[\left\|\sum_{j\in\mathcal{B}_i^{e,t}}\bar{g}_{ij}(\boldsymbol{\theta})\right\|^2\right] - \left\|\sum_{j\in\mathcal{B}_i^{e,t}}G_i(\boldsymbol{\theta})\right\|^2\right) + \frac{pc^2 z_i^2(\epsilon_i, \delta_i, b_i^1, b_i^{>1}, N_i, K, E)}{(b_i^e)^2} \\
&= \frac{1}{(b_i^e)^2}\Big(\underbrace{\mathbb{E}\left[\left\|\sum_{j\in\mathcal{B}_i^{e,t}}\bar{g}_{ij}(\boldsymbol{\theta})\right\|^2\right]}_{\mathcal{A}} - (b_i^e)^2\|G_i(\boldsymbol{\theta})\|^2\Big) + \frac{pc^2 z_i^2(\epsilon_i, \delta_i, b_i^1, b_i^{>1}, N_i, K, E)}{(b_i^e)^2},
\end{aligned}
\tag{13}
$$

where:

$$
\begin{aligned}
\mathcal{A} = \mathbb{E}\left[\left\|\sum_{j\in\mathcal{B}_i^{e,t}}\bar{g}_{ij}(\boldsymbol{\theta})\right\|^2\right] &= \sum_{j\in\mathcal{B}_i^{e,t}}\mathbb{E}\left[\|\bar{g}_{ij}(\boldsymbol{\theta})\|^2\right] + \sum_{m\neq n\in\mathcal{B}_i^{e,t}}2\mathbb{E}\left[[\bar{g}_{im}(\boldsymbol{\theta})]^\top[\bar{g}_{in}(\boldsymbol{\theta})]\right] \\
&= \sum_{j\in\mathcal{B}_i^{e,t}}\mathbb{E}\left[\|\bar{g}_{ij}(\boldsymbol{\theta})\|^2\right] + \sum_{m\neq n\in\mathcal{B}_i^{e,t}}2\mathbb{E}\left[\bar{g}_{im}(\boldsymbol{\theta})\right]^\top\mathbb{E}\left[\bar{g}_{in}(\boldsymbol{\theta})\right] \\
&= b_i^e c^2 + 2\binom{b_i^e}{2}\|G_i(\boldsymbol{\theta})\|^2.
\end{aligned}
\tag{14}
$$

The last equation has used Equation (12) and that we clip the norm of sample gradients $\bar{g}_{ij}(\boldsymbol{\theta})$ with an "effective" clipping threshold $c$. By replacing $\mathcal{A}$ into eq. 13, we can rewrite it as:

$$
\begin{aligned}
\sigma_{i,\tilde{g}}^2(b_i^e) := \mathtt{Var}[\tilde{g}_i^{e,t}(\boldsymbol{\theta})] &= \frac{1}{(b_i^e)^2}\left(\mathbb{E}\left[\left\|\sum_{j\in\mathcal{B}_i^{e,t}}\bar{g}_{ij}(\boldsymbol{\theta})\right\|^2\right] - (b_i^e)^2\left\|G_i(\boldsymbol{\theta})\right\|^2\right) + \frac{pc^2 z_i^2(\epsilon_i,\delta_i,b_i^1,b_i^{>1},N_i,K,E)}{(b_i^e)^2} \\
&= \frac{1}{(b_i^e)^2}\left(b_i^e c^2 + \left(2\binom{b_i^e}{2} - (b_i^e)^2\right)\left\|G_i(\boldsymbol{\theta})\right\|^2\right) + \frac{pc^2 z_i^2(\epsilon_i,\delta_i,b_i^1,b_i^{>1},N_i,K,E)}{(b_i^e)^2} \\
&= \frac{c^2 - \left\|G_i(\boldsymbol{\theta})\right\|^2}{b_i^e} + \frac{pc^2 z_i^2(\epsilon_i,\delta_i,b_i^1,b_i^{>1},N_i,K,E)}{(b_i^e)^2} \approx \frac{pc^2 z_i^2(\epsilon_i,\delta_i,b_i^1,b_i^{>1},N_i,K,E)}{(b_i^e)^2}
\end{aligned}
\tag{15}
$$

The last approximation is valid because $p \gg 1$ ($p$ is the number of model parameters). For instance, $p \approx 2 \times 10^7$ for ResNet-34 for CIFAR100, and $c = 3$, which results in $pz^2(\epsilon_i,\delta_i,q_i,K_i,E)/b_i \gg 1$.

**Scenario 2: the clipping threshold $c$ is ineffective for all samples in a batch:** when the clipping is ineffective for all samples, i.e., $\forall j \in \mathcal{B}_i^{e,t} : c > \|g_{ij}(\boldsymbol{\theta})\|$, we have a noisy version of the batch gradient $g_i^{e,t}(\boldsymbol{\theta}) = \frac{1}{b_i^e}\sum_{j\in\mathcal{B}_i^{e,t}} g_{ij}(\boldsymbol{\theta})$, which is unbiased with variance bounded by $\sigma_{i,g}^2(b_i^e)$ (see Assumption 3.2). We note that $\sigma_{i,g}^2(b_i^e)$ is a constant that depends on the used batch size $b_i^e$. The larger the batch size $b_i^e$ used during round $e$, the smaller the constant. Hence, in this case:

$$
\mathbb{E}[\tilde{g}_i^{e,t}(\boldsymbol{\theta})] = \mathbb{E}[g_i^{e,t}(\boldsymbol{\theta})] = \nabla f_i(\boldsymbol{\theta}),
\tag{16}
$$

and

$$
\begin{aligned}
\sigma_{i,\tilde{g}}^2(b_i^e) = \mathtt{Var}[\tilde{g}_i^{e,t}(\boldsymbol{\theta})] = \mathtt{Var}[g_i^{e,t}(\boldsymbol{\theta})] + \frac{p\sigma_{i,\mathtt{DP}}^2}{(b_i^e)^2} &\leq \sigma_{i,g}^2(b_i^e) + \frac{p\sigma_{i,\mathtt{DP}}^2}{(b_i^e)^2} \\
&= \sigma_{i,g}^2(b_i^e) + \frac{pc^2 z_i^2(\epsilon,\delta,b_i^1,b_i^{>1},N_i,K,E)}{(b_i^e)^2} \\
&\approx \frac{pc^2 z_i^2(\epsilon,\delta,b_i^1,b_i^{>1},N_i,K,E)}{(b_i^e)^2}.
\end{aligned}
\tag{17}
$$

The approximation is again valid because $p \gg 1$ (number of model parameters). Also, note that $\sigma_{i,g}^2(b_i^e)$ decreases with $b_i^e$. Therefore, we got to the same result as in Equation (15).

As observed in Figure 4, $z_i$ grows with $b_i^1$ and $b_i^{>1}$ *sub-linearly* (especially with $b_i^1$). Therefore, the variance of the client $i$'s DP batch gradients $\tilde{g}_i^{e,t}(\boldsymbol{\theta})$ during communication round $e$, decreases with $b_i^e$ fast. The larger the batch size $b_i^e$, the less the noise existing in its batch gradients during the same round.

With the findings above, we now investigate the effect of batch size $b_i^e$ on **the noise level in clients' model updates at the end of round** $e$. During the global communication round $e$, a participating client $i$ performs $E_i^e = K \cdot \lceil \frac{N_i}{b_i^e} \rceil$ batch gradient updates locally with step size $\eta_l$:

$$
\boldsymbol{\theta}_i^{e,k} = \boldsymbol{\theta}_i^{e,k-1} - \eta_l \tilde{g}_i(\boldsymbol{\theta}_i^{e,k-1}), \ \ k = 1,\dots,E_i^e.
\tag{18}
$$

Hence,

$$
\Delta\tilde{\boldsymbol{\theta}}_i^e = \boldsymbol{\theta}_i^{e,E_i^e} - \boldsymbol{\theta}_i^{e,0}
\tag{19}
$$

In each update, it adds a Gaussian noise from $\mathcal{N}(0, \frac{c^2 z_i^2(\epsilon,\delta,b^1,b^{>1},N_i,K,E)}{(b^e)^2}\mathbb{I}_p)$ to its batch gradients independently (see Equation (1)). Hence:

$$
\mathtt{Var}[\Delta\tilde{\boldsymbol{\theta}}_i^e|\boldsymbol{\theta}_i^{e,0}] = E_i^e \cdot \eta_l^2 \cdot \sigma_{i,\tilde{g}}^2(b_i^e),
\tag{20}
$$

where $\sigma_{i,\tilde{g}}^2(b_i^e)$ was computed in Equation (15) and Equation (17), and was a decreasing function of $b_i^e$. Therefore:

$$\text{Var}[\Delta\tilde{\boldsymbol{\theta}}_i^e|\boldsymbol{\theta}_i^{e,0}] \approx K \cdot N_i \cdot \eta_l^2 \cdot \frac{pc^2z_i^2(\epsilon, \delta, b_i^1, b_i^{>1}, N_i, K, E)}{(b_i^e)^3}. \tag{21}$$

$\square$

### E.2 Proof of Lemma 4.2

**Lemma 4.2.** *Let $\Delta_{m,m'}(b^1) := \|\mu_m^*(b^1) - \mu_{m'}^*(b^1)\|$ when $\forall i : b_i^1 = b^1$. The overlap between components $\mathcal{N}\big(\mu_m^*(b^1), \Sigma_m^*(b^1)\big)$ and $\mathcal{N}\big(\mu_{m'}^*(b^1), \Sigma_{m'}^*(b^1)\big)$ is $O_{m,m'} = 2Q(\frac{\sqrt{p}\Delta_{m,m'}(b^1)}{2\sigma^1(b^1)})$, where $(\sigma^1(b^1))^2 := \text{Var}[\Delta\tilde{\boldsymbol{\theta}}_i^1|\boldsymbol{\theta}^{init}, b_i^1 = b^1]$ and $Q(\cdot)$ is the Q function. Furthermore, if we increase $b_i^1 = b^1$ to $b_i^1 = kb^1 \le N$ (for all $i$), we have $O_{m,m'} \le 2Q(\frac{\sqrt{kp}\Delta_{m,m'}(b^1)}{2\rho\sigma^1(b^1)})$, where $1 \le \rho \in \mathcal{O}(1)$ is a small constant.*

*Proof.* We first find the overlap between two arbitrary Gaussian distributions. Without loss of generality, lets assume we are in 1-dimensional space and that we have two Gaussian distributions both with variance $\sigma^2$ and with means $\mu_1 = 0$ and $\mu_2 = \mu$ ($\|\mu_1 - \mu_2\| = \mu$), respectively. Based on symmetry of the distributions, the two components start to overlap at $x = \frac{\mu}{2}$. Hence, we can find the overlap between the two gaussians as follows:

$$O := 2\int_{\frac{\mu}{2}}^{\infty} \frac{1}{\sqrt{2\pi}\sigma}e^{-\frac{x^2}{2\sigma^2}}dx = 2\int_{\frac{\mu}{2\sigma}}^{\infty} \frac{1}{\sqrt{2\pi}}e^{-\frac{x^2}{2}}dx = 2Q(\frac{\mu}{2\sigma}), \tag{22}$$

where $Q(\cdot)$ is the tail distribution function of the standard normal distribution. Now, lets consider the 2-dimensional space, and consider two similar symmetric distributions centered at $\mu_1 = (0,0)$ and $\mu_2 = (\mu, 0)$ ($\|\mu_1 - \mu_2\| = \mu$) and with $\Sigma_1 = \Sigma_2 = \begin{bmatrix} \sigma^2 & 0 \\ 0 & \sigma^2 \end{bmatrix}$. The overlap between the two gaussians can be found as:

$$O = 2\int_{-\infty}^{\infty}\int_{\frac{\mu}{2}}^{\infty} \frac{1}{2\pi\sigma^2}e^{-\frac{x^2+y^2}{2\sigma^2}}dxdy = 2\int_{\frac{\mu}{2}}^{\infty} \frac{1}{\sqrt{2\pi}\sigma}e^{-\frac{x^2}{2\sigma^2}}dx \cdot \int_{-\infty}^{\infty} \frac{1}{\sqrt{2\pi}\sigma}e^{-\frac{y^2}{2\sigma^2}}dy = 2Q(\frac{\mu}{2\sigma}). \tag{23}$$

If we compute the overlap for two similar symmetric $p$-dimensional distributions with $\|\mu_1 - \mu_2\| = \mu$ and variance $\sigma^2$ in every direction, we will get to the same result $2Q(\frac{\mu}{2\sigma})$.

In the lemma, when using batch size $b^1$, we have two Gaussian distributions $\mathcal{N}\big(\mu_m^*(b^1), \Sigma_m^*(b^1)\big)$ and $\mathcal{N}\big(\mu_{m'}^*(b^1), \Sigma_{m'}^*(b^1)\big)$, where

$$\Sigma_m^*(b^1) = \Sigma_{m'}^*(b^1) = \begin{bmatrix} \frac{(\sigma^1(b^1))^2}{p} & & 0 \\ & \ddots & \\ 0 & & \frac{(\sigma^1(b^1))^2}{p} \end{bmatrix}. \tag{24}$$

Therefore, from Equation (23), we can immediately conclude that the overlap between the two Gaussians, which we denote with $O_{m,m'}(b^1)$, is:

$$O_{m,m'}(b^1) = 2Q(\frac{\sqrt{p}\Delta_{m,m'}(b^1)}{2\sigma^1(b^1)}), \tag{25}$$

which proves the first part of the lemma.

Now, lets see the effect of increasing batch size. First, note that we had:

$$\Delta\tilde{\boldsymbol{\theta}}_i^1 = \boldsymbol{\theta}_i^{1,E_i^1} - \boldsymbol{\theta}_i^{1,0},$$
$$\boldsymbol{\theta}_i^{1,k} = \boldsymbol{\theta}_i^{1,k-1} - \eta_l \tilde{g}_i(\boldsymbol{\theta}_i^{1,k-1}), \; k = 1, \ldots, E_i^1, \tag{26}$$

where $E_i^1 = K \cdot \lceil \frac{N}{b^1} \rceil$ is the total number of gradients steps taken by client $i$ during communication round $e = 1$. Therefore, considering that DP batch gradients are clipped with a bound $c$, we have:

$$\|\mathbb{E}[\Delta\tilde{\boldsymbol{\theta}}_i^1(b^1)]\| \leq E_i^1 \cdot \eta_l \cdot c. \tag{27}$$

When we increase batch size $b_i^1$ for all clients from $b^1$ to $kb^1$, the upperbound in Equation (27) gets $k$ times smaller. In fact by doing so, the number of local gradient updates that client $i$ performs during round $e = 1$, which is equal to $E_i^1$, decreases $k$ times. As such, we can write:

$$\Delta\tilde{\boldsymbol{\theta}}_i^1(b^1) = k \cdot \Delta\tilde{\boldsymbol{\theta}}_i^1(kb^1) + \upsilon_i, \tag{28}$$

where $\upsilon_i \in \mathbb{R}^p$ is a vector capturing the discrepancies between $\Delta\tilde{\boldsymbol{\theta}}_i^1(b^1)$ and $k \cdot \Delta\tilde{\boldsymbol{\theta}}_i^1(kb^1)$. Therefore, we have:

$$\mu_m^*(b^1) = \mathbb{E}[\Delta\tilde{\boldsymbol{\theta}}_i^1(b^1)|s(i) = m] = \mathbb{E}[k \cdot \Delta\tilde{\boldsymbol{\theta}}_i^1(kb^1) + \upsilon_i|s(i) = m]$$
$$= k \cdot \mathbb{E}[\Delta\tilde{\boldsymbol{\theta}}_i^1(kb^1)] + \mathbb{E}[\upsilon_i|s(i) = m] = k \cdot \mu_m^*(kb^1) + \mathbb{E}[\upsilon_i|s(i) = m]. \tag{29}$$

Therefore, we have:

$$\|\mu_m^*(b^1) - \mu_{m'}^*(b^1)\| = \left\| k\mu_m^*(kb^1) - k\mu_{m'}^*(kb^1) + \left( \mathbb{E}[\upsilon_i|s(i) = m] - \mathbb{E}[\upsilon_i|s(i) = m'] \right) \right\|. \tag{30}$$

Based on our experiments, the last term above, in parenthesis, is small and we can have the following approximation for the equation above:

$$\|\mu_m^*(b^1) - \mu_{m'}^*(b^1)\| \approx \|k\mu_m^*(kb^1) - k\mu_{m'}^*(kb^1)\|, \tag{31}$$

or equivalently:

$$\|\mu_m^*(kb^1) - \mu_{m'}^*(kb^1)\| \approx \frac{\|\mu_m^*(b^1) - \mu_{m'}^*(b^1)\|}{k}. \tag{32}$$

Figure 14 (left) shows the validity of the approximation above with some experimental results. On the other hand, from Equation (2) and also noting that a client, with dataset size $N$ and batch size $b^1$, takes $\frac{N}{b^1}$ gradient steps during each epoch of the first round, we have:

$$\forall m \in [M] : (\sigma_m^1(b^1))^2 = (\sigma^1(b^1))^2 \approx K \cdot N \cdot \eta_l^2 \cdot \frac{pc^2 z^2(\epsilon, \delta, b^1, b^{>1}, N, K, E)}{(b^1)^3}. \tag{33}$$

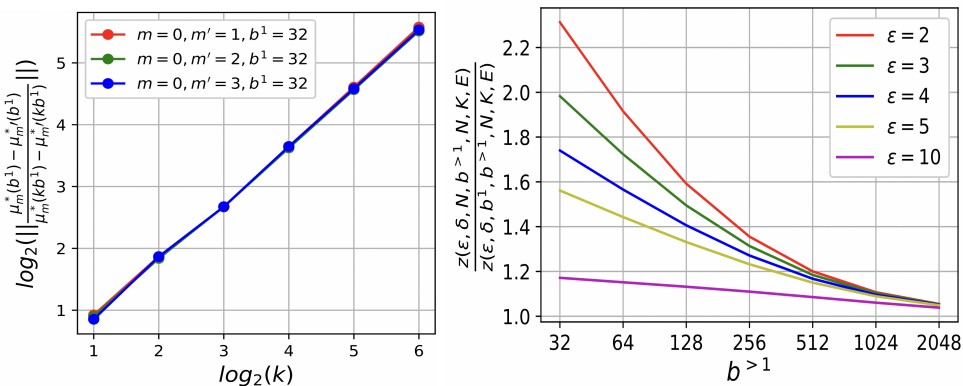

Figure 14: **Left:** Distance between the centers of different clusters, i.e., the distance between $\mu_m^*(b^1)$ and $\mu_{m'}^*(b^1)$, decreases $k$ times as $b^1$ increases $k$ times. The three curves in the plot are obtained on CIFAR10 with 4 clusters $m \in \{0, 1, 2, 3\}$ obtained from covariate shift (rotation). The curves are overlapping all with slope 0.95, which is very close to 1. This shows the validity of the approximation in Equation (32). **Right:** Effect of changing batch size $b^1$ to full batch size in the first round on the noise scale $z$. In the denominator, $b^1$ is equal to $b^{>1}$. Results are obtained from Rényi-DP accountant (Mironov et al., 2019) with $N = 50000$, $K = 1$ and $E = 200$. For each value of $\epsilon$, we have shown the results for seven values of $b^{>1}$.

When we change the batch size used during the first communication round $e = 1$ from $b^1$ to $kb^1$ and we fix the batch size of rounds $e > 1$, then the noise scale $z$ changes from $z(\epsilon, \delta, b^1, b^{>1}, N_i, K, E)$ to $z(\epsilon, \delta, kb^1, b^{>1}, N_i, K, E)$. Confirmed by our experimental analysis (see Figure 14, right), the amount of change in $z$ due to this is small, as we have changed the batch size only in the first round $e = 1$ from $b^1$ to $kb^1$, while the batch sizes in the other $E - 1$ rounds are unchanged and $E \gg 1$. Therefore, supported by the results in Figure 14, we can always establish an upper bound on the amount of change in $z$ as $b^1$ increases: $z(\epsilon, \delta, kb^1, b^{>1}, N, K, E) \leq \rho z(\epsilon, \delta, b^1, b^{>1}, N, K, E)$, where $\rho$ is a small constant (e.g., $\rho = 2.5$ in Figure 14). So we have:

$$
\begin{aligned}
\forall m \in [M] : (\sigma_m^1(kb^1))^2 = (\sigma^1(kb^1))^2 &\approx K \cdot N \cdot \eta_l^2 \cdot \frac{pc^2 z^2(\epsilon, \delta, kb^1, b^{>1}, N, K, E)}{(kb^1)^3} \\
&\leq K \cdot N \cdot \eta_l^2 \cdot \frac{pc^2 \rho^2 z^2(\epsilon, \delta, b^1, b^{>1}, N, K, E)}{(kb^1)^3} \\
&= \frac{\rho^2 (\sigma^1(b^1))^2}{k^3}.
\end{aligned}
\tag{34}
$$

From Equation (32) and Equation (34), we have:

$$
O_{m,m'}(kb^1) = 2Q\left( \frac{\sqrt{p}\Delta_{m,m'}(kb^1)}{2\sigma(kb^1)} \right) \leq 2Q\left( \frac{\sqrt{p}\frac{\Delta_{m,m'}(b^1)}{k}}{2\frac{\rho\sigma(b^1)}{k^{\frac{3}{2}}}} \right) = 2Q(\frac{\sqrt{kp}\Delta_{m,m'}(b^1)}{2\rho\sigma(b^1)}),
\tag{35}
$$

which completes the proof. $\qquad\square$

### E.3 Proof of Theorem 4.3

**Theorem 4.3.** *(Ma et al., 2000) Given model updates $\{\Delta\tilde{\boldsymbol{\theta}}_i^1(b^1)\}_{i=1}^n$, as samples from a true mixture of Gaussians $\psi^*(b^1) = \{\mathcal{N}\left(\mu_m^*(b^1), \Sigma_m^*(b^1)\right), \alpha_m^*\}_{m=1}^M$, if $O^{max}(\psi^*(b^1))$ is small enough, then:*

$$\lim_{r \to \infty} \frac{\|\psi^{r+1} - \psi^*(b^1)\|}{\|\psi^r - \psi^*(b^1)\|} = o\left(\left[O^{max}(\psi^*(b^1))\right]^{0.5-\gamma}\right), \tag{5}$$

*as $n$ increases. $\psi^r$ is the `GMM` parameters returned by `EM` after $r$ iterations. $\gamma$ is an arbitrary small positive number, and $o(x)$ means it is a higher order infinitesimal as $x \to 0 : \lim_{x \to 0} \frac{o(x)}{x} = 0$.*

*Proof.* The proof directly follows from the proof of Theorem 1 in Ma et al. (2000) by considering $\{\Delta\tilde{\boldsymbol{\theta}}_i^1(b^1)\}_{i=1}^n$ as the samples of Gaussian mixture $\{\mathcal{N}(\mu_m^*(b^1), \Sigma_m^*(b^1)), \alpha_m^*\}_{m=1}^M$. □

### E.4 Proof of Theorem 5.1

**Theorem 5.1.** *The set of model updates $\{\Delta\tilde{\boldsymbol{\theta}}_i^e\}_{e=1}^E$, which are uploaded to the server by each client $i \in \{1, \cdots, n\}$ during the training time, as well as their private local model cluster selections satisfy $(\epsilon, \delta)$-DP, where the parameters $\epsilon$ and $\delta$ depend on the `DP` noise variance $\sigma_{i,DP}^2$ used by the client for `DPSGD` (Equation (1)) and the parameter $\epsilon_{select}$ used for its private cluster selections using exponential mechanism (Equation (8)).*

*Proof.* The sensitivity of the batch gradient in Equation (1) to every data sample is $c$. Therefore, based on Proposition B.4, each of the batch gradient computations by client $i$ (in the first round $e = 1$ as well as the next rounds $e > 1$) is $(\alpha, \frac{\alpha c^2}{2\sigma_{i,DP}^2})$-RDP. Therefore, if the client runs $E_i^{\text{tot}}$ total number of gradient updates during the training time, which results in the model updates $\{\Delta\tilde{\boldsymbol{\theta}}_i^e\}_{e=1}^E$ uploaded to the server, the set of model updates will be $(\alpha, \frac{E_i^{\text{tot}}\alpha c^2}{2\sigma_{i,DP}^2})$-RDP, according to Theorem B.2. Finally, according to Lemma B.3, this guarantee is equivalent to $(\frac{E_i^{\text{tot}}\alpha c^2}{2\sigma_{i,DP}^2} + \frac{log(1/\delta)}{\alpha-1}, \delta)$-DP (for any $\delta > 1$). The `RDP`-based guarantee can be computed over a bunch of orders $\alpha$ and the best result among them is chosen as the privacy guarantee. Therefore, the set $\{\Delta\tilde{\boldsymbol{\theta}}_i^e\}_{e=1}^E$ satisfies $(\epsilon, \delta)$-DP, with $\epsilon = \frac{E_i^{\text{tot}}\alpha c^2}{2\sigma_{i,DP}^2} + \frac{log(1/\delta)}{\alpha-1}$ derived above, and $\delta > 0$. On the other hand, clients' local cluster selections are also privatized by exponential mechanism and satisfy $(\epsilon, \delta)$-DP. Therefore, the overall training process for each client is private and satisfies $(\epsilon, \delta)$-DP. Tighter bounds for $\epsilon$ can be derived by using the numerical procedure, proposed in (Mironov et al., 2019), for accounting sampled Gaussian mechanism. □

## F The relation between Lemma 4.1 and the law of large numbers

We first state the weak law of large numbers and then explain how Lemma 4.1 is closely related to it.

**Theorem F.1** (Weak law of large numbers (Billingsley, 1995)). *Suppose that $\{X_i\}_{i=1}^b$, is an independent sequence (of size $b$) of i.i.d random variables with expected value $\mu$ and positive variance $\sigma^2$. Define $\bar{X}_b = \frac{\sum_{i=1}^b X_i}{b}$ as their sample mean. Then, for any positive number $\Delta > 0$:*

$$\lim_{b \to \infty} Pr[|\bar{X}_b - \mu| > \Delta] = 0. \tag{36}$$

In fact, the weak law of large numbers states that the sample mean of some i.i.d random variables converges in probability to their expected value ($\mu$). Furthermore, we can see that $\text{Var}[\bar{X}_b] = \frac{\sigma^2}{b}$, which means that *the variance of the sample mean decreases as the sample size $b$ increases.*

Now, remember from Equation (1) that when computing the `DP` stochastic batch gradients in round $e$ (with batch size $b_i^e$), we add `DP` noise with variance $\sigma_{i,DP}^2/b_i^e$ to each of the $b_i^e$ clipped sample gradients in the batch *and average the resulting $b_i^e$ noisy clipped sample gradients.* The sampled noise terms added to the clipped sample gradients in a batch are i.i.d with mean zero. Therefore, based on the above theorem, the variance of their average over each batch should approach zero as the batch size $b_i^e$ grows. The same discussion applies to all the $K \cdot N_i/b_i^e$ gradient updates performed by client $i$ during a communication round $e$ (whose noises will be summed up), which results in Lemma 4.1.

## G    Gradient accumulation

When training large models with `DPSGD`, increasing the batch size results in memory exploding during training or finetuning. This might happen even when using ordinary SGD. On the other hand, using a small batch size results in larger stochastic noise in batch gradients. Also, in the case of `DP` training, using a small batch size results in fast increment of `DP` noise (as explained in Lemma 4.1 in details). Therefore, if the memory budget of devices allow, we prefer to avoid using small batch sizes. But what if there is a limited memory budget? A solution for virtually increasing batch size is "gradient accumulation", which is very useful when the available physical GPU memory is insufficient to accommodate the desired batch size. In ordinary SGD, we set the gradient w.r.t every parameter to zero at the beginning of each back propagation and compute the gradient w.r.t each parameter. Finally, the computed batch gradients are used for updating the parameters at the end of the back propagation. In contrast to this, gradient accumulation accumulates gradients w.r.t parameters over multiple smaller batches. When the accumulated gradients reach the target logical batch size, the model weights are updated with the accumulated batch gradients. This method simulates a large batch size with zero overheads. The page in `https://opacus.ai/api/batch_memory_manager.html` explains more details.

## H    Can clients use data augmentation in the first round to simulate a large dataset size?

Based on the findings in Section 4.2, and noting that $b_i^1 \leq N_i$, a question about R-DPCFL is that can the clients use data augmentation in the first round to simulate a large dataset size and consequently improve the upperbound $N_i$ on $b_i^1$? The short answer is negative, due to potential privacy leakage caused by data augmentation. Let us focus on a client $i$ and let $u = (x, y) \in \mathcal{D}_i$ be a train "sample" in its local data. Let $\tau$ be the set of all possible transformations and let $T(u) = \{t(u), t \in \tau\}$ be the set of all augmented "instances" of the sample $u$. Hence, after augmentation, we get the $|\tau|$ times larger augmented dataset $\mathcal{D}_{i,aug} = \cup_{u \in \mathcal{D}_i} T(u)$. $\mathcal{D}_{i,aug}$ makes the upper bound on batch size $b_i^1$, $|\tau|$ times larger, which seems desirable. However, the augmentation leads to an extra privacy leakage for the client, as explained below.

Let us first assume client $i$ uses batch size $b_i^1$ and no data augmentation in the first round. Remembering Equation (1), at the gradient step $t$ during the first round, it computes the noisy clipped batch gradient

$$\tilde{g}_i^{1,t}(\boldsymbol{\theta}) = \frac{1}{b_i^1}\Big[\Big(\sum_{u \in \mathcal{B}_i^{1,t}} \bar{g}_{iu}(\boldsymbol{\theta})\Big) + \mathcal{N}(0, \sigma_{i,\text{DP}}^2 \mathbb{I}_p)\Big], \tag{37}$$

where the batch $\mathcal{B}_i^{1,t}$ has size $b_i^1$ and also $\sigma_{i,\text{DP}} = c \cdot z_i(\epsilon, \delta, b_i^1, b_i^{>1}, N_i, K, E)$ is fixed. In the literature, there have been some methods for "differentially private data augmentation" (Hoffer et al., 2019; De et al., 2022). These methods propose to use "batch augmentation" by replicating *the samples present in a batch* with their augmented instances (e.g., replicating $u \in \mathcal{B}_i^{1,t}$ with $T(u)$). Furthermore, the gradients across the augmented instances $T(u)$ of a sample $u \in \mathcal{B}_i^{1,t}$ are clipped and averaged (De et al., 2022) *before* the `DP` noise addition. More specifically, instead of the noise batch gradient above, the following noisy batch gradient is computed for `DP` batch augmentation:

$$\tilde{g}_i^{1,t}(\boldsymbol{\theta}) = \frac{1}{b_i^1}\Big[\sum_{u \in \mathcal{B}_i^{1,t}} \Big(\frac{1}{\tau}\sum_{j \in T(u)} \bar{g}_{ij}(\boldsymbol{\theta})\Big) + \mathcal{N}(0, \sigma_{i,\text{DP}}^2 \mathbb{I}_p)\Big]. \tag{38}$$

In this way, the sensitivity of the batch gradient "to each sample $u \in \mathcal{B}_i^{1,t}$" does not change. So when we use the same noise variance $\sigma_{i,\text{DP}}^2$ as before augmentation, we still get the same `DP` privacy guarantee w.r.t each sample $u$ in the dataset $\mathcal{D}_i$.

In contrast to the batch augmentation discussed above, using "dataset augmentation" and full batch size simultaneously in the first round leads to an extra privacy leakage, *if client $i$ keeps using the same DP noise variance $\sigma_{i,DP}^2$*. In this case, $b_i^1 = \tau N_i$, and the single noisy gradient in the first round can be written as:

$$\tilde{g}_i^{1,t}(\boldsymbol{\theta}) = \frac{1}{\tau N_i}\Big[\Big(\sum_{j\in\mathcal{D}_{i,aug}}\bar{g}_{ij}(\boldsymbol{\theta})\Big) + \mathcal{N}(0,\sigma_{i,\mathtt{DP}}^2\mathbb{I}_p)\Big] = \frac{1}{N_i}\Big[\sum_{u\in\mathcal{D}_i}\Big(\frac{1}{\tau}\sum_{j\in T(u)}\bar{g}_{ij}(\boldsymbol{\theta})\Big) + \mathcal{N}(0,\frac{\sigma_{i,\mathtt{DP}}^2}{\tau^2}\mathbb{I}_p)\Big], \qquad (39)$$

which is the same private "batch augmentation" in Equation (38) with $b_i^1 = N_i$ (i.e. full batch size) **but with $\tau^2$ times smaller effective $\mathtt{DP}$ noise variance**. This means that the privacy guarantee that we get after using "dataset augmentation" and full batch size is not the same as the guarantee for using only full batch size (with the actual dataset size). Therefore, one should be cautious when using dataset augmentation in $\mathtt{DP}$ settings.

In the following, we have also mentioned the recent findings about the effect of data augmentation on membership inference attacks ($\mathtt{MIA}$) to $\mathtt{ML}$ models, which generally supports the discussion above from another point of view.

It is widely believed that the capacity of $\mathtt{MIA}$ (Shokri et al., 2017; Yeom et al., 2017; Salem et al., 2019; Jia et al., 2019; Zarifzadeh et al., 2024) to $\mathtt{ML}$ models is largely attributed to the models' generalization gap: the difference between their average loss on train and test sets (Shokri et al., 2017; Yeom et al., 2017; Li et al., 2020a). Searching for the root causes of model's vulnerability to $\mathtt{MIA}$, some works proposed overfitting as one factor (Yeom et al., 2017). Following this, some works suggested that data augmentation, as well as improving the utility, may also reduce the privacy risk to $\mathtt{MIA}$ by mitigating overfitting to train data (Shokri et al., 2017; Sablayrolles et al., 2019). However, the extensive experimental results in (Kaya & Dumitras, 2021) shows that when data augmentation is applied with low-intensity for boosting the model's accuracy, it fails to achieve substantial protection against $\mathtt{MIA}$. On the other hand, high intensity augmentation, e.g., cropping 90% of an image, which often results in a utility drop, reduces the privacy risk of the model and its vulnerability to $\mathtt{MIA}$. These findings suggest that a smaller generalization gap, which is usually provided by low-intensity data augmentation, does not necessarily translate to a lower privacy risk. This along with the findings in (Shokri et al., 2017; Yeom et al., 2017; Li et al., 2020a) suggests that overfitting might be a sufficient but not necessary condition for the success of $\mathtt{MIA}$.

Similar to Kaya & Dumitras (2021), the work in (Yu et al., 2021) challenges the observations in (Shokri et al., 2017; Sablayrolles et al., 2019) that data augmentation improves the vulnerability to $\mathtt{MIA}$. The work considers $\epsilon$-$\mathtt{DP}$ definition and shows that when the attacker has access to one "single instance" in $T(u)$, where $u$ is a data sample and $T(u)$ is its all augmented instances, its ability for inferring membership of $u$ decreases with data augmentation (this is in contrast with some of the experimental results in (Kaya & Dumitras, 2021)). On the other hand, when the attacker has access to all the augmented instances of $u$ (i.e., it has access to all the instances in $T(u)$), the situation is different. In this case, they formulate $\mathtt{MIA}$ as a "set classification" problem to classify the set of augmented instances $T(u)$ as being used during training or not. They show that, this approach can infer the membership of $T(u)$ with more success rate, compared to the attacks which have access to only one instance in $T(u)$. Whether an attacker has access to all the augmented instances in $T(u)$ for a sample $u$ or not depends on the problem in hand and the considered security model. Yet, the findings by Yu et al. (2021) suggests that *$\mathtt{MIA}$ to models trained with data augmentation could be largely underestimated, if the attacker has access to a single instance at the attack time.*

## I  Further related works

**Performance parity in $\mathtt{FL}$**: Performance parity of the final trained model across clients is an important goal in $\mathtt{FL}$. Addressing this goal, Mohri et al. (2019) proposed Agnostic $\mathtt{FL}$ ($\mathtt{AFL}$) by using a min-max optimization approach. $\mathtt{TERM}$ (Li et al., 2020b) used tilted losses to up-weight clients with large losses. Finally, Li et al. (2020c) and Zhang et al. (2023) proposed $q$-$\mathtt{FFL}$ and $\mathtt{PropFair}$, inspired by $\alpha$-fairness (Lan et al., 2010) and proportional fairness (Bertsimas et al., 2011), respectively. Generating one common model for all clients, these techniques do not perform well when the data distribution across clients is highly heterogeneous or a structured data heterogeneity exists across clusters of clients. While model personalization techniques (e.g., $\mathtt{MR}$-$\mathtt{MTL}$ (Liu et al., 2022a)) are proposed for the former case, stronger personalization techniques, e.g., client clustering, are used for the latter.

**Differential privacy, group fairness and performance parity:** Gradient clipping and random noise addition used in `DPSGD` disproportionately affect underrepresented groups. Some works tried to address the tension between group fairness and `DP` in centralized settings (Tran et al., 2020) (by using Lagrangian duality) and `FL` settings (Pentyala et al., 2022) (by using Secure Multiparty Computation (`MPC`)). Another work tried to remove the disparate impact of `DP` on model performance of minority groups in centralized settings (Esipova et al., 2023), by preventing gradient misalignment across different groups of data. Unlike the previous works on group fairness, our work adopts cross-model fairness, where the utility drop after adding `DP` must be close for different groups (Dwork et al., 2012), including minority and majority clients. As we consider a structured data heterogeneity across clients, the mentioned approaches are not appropriate, due to generating one single model for all.