# OpenReview forum: "Differentially Private Clustered Federated Learning"
_TMLR — Accepted by TMLR_

### Review · Reviewer_SLhf · 2025-06-21

**Summary Of Contributions:**

The paper tackles the problem of sample-level differentially private federated learning under structured data heterogeneity. It introduces R-DPCFL, a three-stage clustered FL algorithm that (i) uses full-batch local training in the first round to obtain low-noise model updates, (ii) performs soft clustering of clients with a GMM whose confidence is quantified by a minimum separation score, and (iii) gradually switches to loss-based private clustering via the exponential mechanism.

**Audience:**

Yes

**Broader Impact Concerns:**

No concerns. This is a theoretical work that may have societal impacts in the future.

**Claims And Evidence:**

Yes

**Requested Changes:**

I think it would be beneficial if some of the following points are addressed:
- Include an ablation where some clients opt out of the full-batch round to test robustness.
- Report results on at least one non-vision dataset to demonstrate broader applicability.
- Compare with recent personalized DP-FL methods that incorporate clustering implicitly (e.g., FedMix/FedNova variants).
- Comment on the correctness of theory given an mis-specified $M$.

**Strengths And Weaknesses:**

Strengths:

- The algorithmic design is well-motivated. The full-batch first round combined with hybrid update clustering directly addresses the brittleness of previous loss-only or gradient-only clustered FL under DP noise.
- Sound theory that links batch size, DP noise, and EM convergence, providing intuition that is immediately actionable for practitioners.
- Thorough evaluation with four datasets, multiple heterogeneity types, strong baselines, per-cluster analysis, and ablations on batch size, $\varepsilon$, dataset size, and hyperparameters.


Weaknesses:


- All experiments use image classification; evaluation on tabular tasks would test generality.
- Privacy budgets. Main results use rather loose $\varepsilon \in \{5, 10\}$; additional runs at tighter $\varepsilon \leq 2$ would contextualize utility–privacy trade-offs.
- The theory might not hold if the number of cluster is mis-specified.
- Gaussian-mixture separability and homogeneous variance may not hold in practice.

---

> ### Author Response · Authors · 2025-07-12
> **Official Comment**
>
> We appreciate the reviewer for the constructive comments and questions. We are happy to answer them in the the following:
>
> > comment1: All experiments use image classification.
>
> Using image classification datasets are very common in the literature of clustered FL [1,2,3]. The reason is that for images, we can easily simulate covariate shift (by rotation) and concept shift (by label flipping). While we have not seen any works using tabular data for CFL, we are open to  tabular datasets suggestions that could be appropriate for this context.
>
> [1] M. Werner, et. al., Provably Personalized and Robust Federated Learning, TMLR 2023.
>
> [2] Z. Liu, et. al., On Privacy and Personalization in Cross-Silo Federated Learning, Neurips 2022.
>
> [3] A. Ghosh, et. al., An Efficient Framework for Clustered Federated Learning, 2021.
>
> > comment2:  Main results use rather loose $\epsilon \in \\{5, 10 \\}$.
>
> The main results are reported in figures 6, 7 and 9 and figures 11 and 12 in the appendix, **which are for all values of $\epsilon$ in the set $\\{3,4,5,10 \\}$**.  To fully address the comment, the following results are on MNIST with covariate shift and $\epsilon=2$, complementing the results in Fig. 6, left.
>
> * MNIST ($\epsilon=2$):
> |Algorithm| R-DPCFL | MR-MTL | Local| IFCA|
> | --------| --------| --------|--------| -------- |
> |Average accuracy            |89.12             |86.11         |85.21      | 84.11|
> |Clustering accuracy       | 4/4                 |  -                | -                |1/4|
>
>
>
> > comment 3: The theory might not hold if the number of clusters is mis-specified.
>
> Lemma 4.2 and theorem 4.3  depend on the "true cluster centres" ($\mu_m^*$), and accordingly, they have not been meant and should not be expected to work for an estimated $M$. In fact, their main message is: "by using a large batch size in the first round it will be much easier to "see" the underlying cluster structure of clients". **Similarly, it will be easier to “estimate” the true number of clusters when it is unknown** (as we observed in appendix C.9 and figure 10).
>
> > comment 4: Gaussian-mixture separability and homogeneous variance may not hold in practice.
>
> In this case, the value of $MPO$ (explained in section 4.3) will be close to 1, and the switching time $E_c = (1-MPO)E/2$ will be sooner, and R-DPCFL will be "soft clustering” clients between rounds 1 and $E_c$ and "hard clusters” them afterwards, while the existing baselines (IFCA) will be “hard clustering” clients as before from the very first round.
>
> Furthermore, whatever DP noise variance that a client has, if it uses a full batch size in the first round, the amount of DP noise in its model update will be minimized. Small deviations in the values of the noise variances of clients will result in the clients' model updates being scattered around their corresponding  true cluster centres ($\mu_m^*$), similar to figure 5, right.
>
>
>
>  > RCH1: partial participation.
>
> The following table shows the results for a participation rate of 80% for clients in the first round, on CIFAR10 with covariate shift and $\epsilon=5$ and averaged over 4 trials. The results in the third column of Fig.6 were the corresponding results with full client participation.
>
> |Algorithm| R-DPCFL | MR-MTL | Local| IFCA|
> | --------| --------| --------|--------| -------- |
> |Average accuracy            |73.81             |70.82         |71.23      | 68.42|
> |Clustering accuracy       | 4/4                 |  -                | -                |2/4|
>
> > RCH2:
>
> commented on this above.
>
> > RCH3: Personalized DP-FL methods.
>
> The FedMix algorithm [1] is a personalization method performing a tuneable personalization between the global model and the clients’ local models, as observed in equations 1 and 2 in [1]. **We have already used such an algorithm with implicit clustering and personalization** as one of our baselines. In fact, the baseline “MR-MTL” is a personalization method, which we have explained about in sections C.3 and C.4 of the appendix. However, due to the high data heterogeneity existing across clusters in clustered FL settings, this category of methods cannot outperform R-DPCFL, as we observe in Fig. 6 and 7 clearly.
>
> [1] E. Gasanov, et. al., “FedMix: A Simple and Communication-Efficient Alternative to Local Methods in Federated Learning”, 2021
> https://openreview.net/pdf?id=6WhWweNk8gr
>
> Finally, the FedNoVa algorithm [2] is neither a personalization nor a clustering algorithm. It aims at eliminating the potential heterogeneity existing in “clients’ dataset sizes” by the server aggregating clients’ local model updates normalized by their corresponding number of local updates ($E_i = K \times \frac{N_i}{b_i}$).
>
> [2] J. Wang, et. al., “Tackling the Objective Inconsistency Problem in Heterogeneous Federated Optimization”, 2020
> https://arxiv.org/pdf/2007.07481
>
>
> > RCH4: Comment on the correctness of theory given an mis-specified $M$.
>
> We refer the reviewer to our response to comment 3.

---

### Review · Reviewer_B8k9 · 2025-07-07

**Summary Of Contributions:**

The paper proposes DP clustered FL algorithm for heterogeneous data distributions referred to as R-DPCFL. The proposed method aims to address two major vulnerabilities in the clustering process of existing DP clustered FL algorithms namely 1) sensitivity to model initialization, and 2) sensitivity to randomness in clients’ model updates due to stochastic noise. R-DPCFL has three stages of training:
1. GMM with M components: In this stage, all the clients run one step of DP-SGD with full batch which results in less noisy (stochastic noise). These model updates are then used to learn the cluster centers of a GMM model with M components. The number of clusters (M) is either given or can be found by maximizing the confidence of the learned GMM.
2. Soft clustering: This stage is run for $E_c$ number of epochs. The server determines the cluster of each client based on $\pi_i[m]$ determined in the previous stage where $\pi_i[m]$ is the probability of client $i$ belonging to cluster $m$. Each client contributes to the chosen cluster via DP-SGD model update. Server updates each cluster's model via weighted aggregation of the model updates from the clients belonging to the given cluster.
3. local hard clustering: In this stage ($E_c to E$), server sends all the models to each client and the client picks a cluster/model that results in minimum train loss/accuracy. The clients select the cluster centers in a private way using exponential mechanism.

The  paper presents theoretical results showing the noise level in the DP-SGD model updates decreases with increase in batch-size. The paper also establishes that the computational complexity of learning the GMM in the first round also decreases fast as batch-size increases.

The experimental results on MNIST, FMNIST and CIFAR-10 with a setup of four clusters of clients indexed by m ∈ {0, 1, 2, 3} with {3, 6, 6, 6} clients show that the proposed algorithm performs better than three chosen baselines (IFCA: clustering based on local train loss/accuracy, local training: clients do not participate FL and learn a local model by running DPSGD on their local data, and MT-MRL: learns a personalized model for each client while regularizing the distance from the averaged model across all clients). The experiments focus on covariate and concept shifts to simulate  heterogeneous data distributions. simple 2-layer CNN is used for MNIST/FMNIST training and ResNet-18 for CIFAR-10.

**Audience:**

Yes

**Claims And Evidence:**

Yes

**Requested Changes:**

1. What is the compute and communication overhead of the proposed R-DPCFL as compared to DP extension of IFCA?
2. What are the limitations of the proposed method? Please add a section on limitation either in the main paper or appendix.
3. What is the impact on performance (test loss/accuracy) when there is no stage 3 ie $E_c=E$?

**Strengths And Weaknesses:**

Strengths:
1. The paper address an important problem in Federated Learning ie Data Heterogeneity.
2. The proposed algorithm uses both model updates and training loss/accuracy values of clients, and mitigating noise impacts with large initial batch sizes to enhance the clustering accuracy.
3. The paper presents theoretical results indicating the positive impact of  large batch sizes on the clustering accuracy.
4. Experimental results presented on 3 different (small) datasets show that the proposed method performs better than the baselines.

Weaknesses/questions:
1. The paper uses sample-level DP. However, for FL use-cases user-level DP is more realistic.
2. The paper doesn't mention anything about participation rate. Did the paper assume full participation rate? If yes, then can the authors talk about the generalization properties of the algorithm to a generic case with partial client participation?
3. The third stage is very communication heavy as the server needs to communicate M models to all the clients.
4. The paper doesn't present any theoretical analysis on the convergence rate of the proposed algorithm.
5. Bar plots in Figure 6 and 7 are not clear to me. What does 10% for IFCA mean? Is full batch-size the main reason why R-DPCFL always determines the right clusters?

---

> ### Author Response · Authors · 2025-07-12
> **Official Comment**
>
> We appreciate the reviewer for the constructive comments and questions. We are happy to answer them in the the following:
>
> > Comment 1: sample-level DP and user-level DP.
>
> Client-level DP may not be suitable for cross-silo FL, where there are fewer clients but each hold many data subjects that require protection. For example, when hospitals/banks/schools wish to federate patient/customer/student records, **it is the people owning those records rather than the participating silos that should be protected** [1].
>
> [1] Z. Liu, et. al., On Privacy and Personalization in Cross-Silo Federated Learning, Neurips 2022.
>
> > Comment 2: Partial client participation.
>
> The R-DPCFL algorithm can be used for partial client participation as well. The following table shows the results for a participation rate of 80% for clients in the first round, on CIFAR10 with covariate shift and $\epsilon=5$ and averaged over 4 runs. The results in the third column of Fig.6 are the corresponding results with full client participation.
> |Algorithm| R-DPCFL | MR-MTL | Local| IFCA|
> | --------| --------| --------|--------| -------- |
> |Average accuracy            |73.81             |70.82         |71.23      | 68.42|
> |Clustering accuracy       | 4/4                 |  -                | -                |2/4|
>
> > Comment 3: The third stage is very communication heavy.
>
> That after round $E_c$ the server sends $M$ models to the clients and they choose one locally **is not specific to R-DPCFL**. In fact, IFCA does so from “the very first round”. Second, we have reduced this overload a lot, as explained in section C.6 of the appendix. More specifically, we let clients run local clustering for only 10% of the total number of rounds $E$. For example, IFCA runs local clustering during the first $ ⌊E/10⌋$ rounds, **and fixes clients’ cluster assignments afterwards**. Similarly, R-DPCFL asks clients run local clustering during rounds $E_c ≤e≤ E_c + ⌊E/10⌋$, **and fixes clients’ cluster assignments afterwards**. This is possible because during the first $E_c$ rounds, the $M$ cluster models have been trained enough and only one of them is the best for each client **with a high margin**. Therefore, the same model would be selected over and over again if we had continued the local clustering for all the rounds after $E_c$. **The 10% that you see in Figures 6 and 7 legends and captions also refer to this point**. Therefore, we have effectively addressed this comment in the current draft.
>
> > Comment 4: Theoretical analysis on the convergence rate.
>
> We have provided theoretical results in Theorem 4.3 about the convergence rate of the GMM learning at the end of the first round. A theoretical analysis of the overall convergence of the algorithm is complicated and will need to consider the potential clustering errors that may happen during training time, and was  not the focus of our work. The main message of our paper was to show the overlooked effect of large batch sizes in the clustered FL literature and how effectively it can improve the clustering accuracy.
>
> > Comment 5: Is full batch size the reason ...?
>
> We answered this question above in “Comment 3”. Also, the main reason behind the success rate of R-DPCFL is the usage of the full batch size in the first round and heavily reducing the noise level in the clients’ local model updates at the end of the round. While R-DPCFL operates on this much less noisy model updates, IFCA operates completely randomly by a random initialization of clusters’ models and clustering clients based on their loss values. Hence, there is no guarantee or even expectation for IFCA to find the true clusters.
>
> > Q1: What is the compute and communication overhead of the proposed R-DPCFL as compared to DP extension of IFCA?
>
> The only computational overhead of R-DPCFL compared to the DP extension of IFCA is that it learns a GMM at the end of the very first round. Importantly, as we proved in Theorem 4.3 and Fig. 9 left, this overhead drops very quickly by using large batch sizes $\\{b_i^1\\}_{i=1}^n$ in the very first round, as R-DPCFL indeed does. Also, for simulating a large batch size in the first round, we use the gradient accumulation technique explained in section G of the appendix, which has no overheads. Therefore, R-DPCFL has an indeed negligible overhead compared to IFCA.
>
> > Q2: What are the limitations of the proposed method?
>
> We have explained the limitations of the work in the conclusion. R-DPCFL needs the clients to have large dataset sizes so that using a full batch size by clients in the first round would be effective in reducing the noise level in clients' model updates at the end of the first round. This is usually the case in cross-silo FL settings (e.g. banks, hospitals, schools) .
>
>  > Q3: What is the impact when $E_c=E$?
>
> This never happens. As we have explained in the last paragraph of page 11, $E_c$ can be at maximum equal to $E/2$, and after that the clustering strategy will be switched to loss-based clustering.

---

### Review · Reviewer_4b8Q · 2025-07-07

**Summary Of Contributions:**

Existing clustered Federated Learning (FL) methods in differentially private FL (DPFL) under high structured data heterogeneity are sensitive to DP noise. This paper proposes a novel algorithm for differentially private clustered FL, which is robust to DP noise and accurately identifies underlying client clusters. Key contributions include:

1. Clustering clients based on both their model updates and training loss values. Mitigating server uncertainties when clustering clients' model updates at the first round's end by using large batch sizes and Gaussian Mixture Models (GMM), reducing the impact of DP and stochastic noise to avoid clustering errors.
2. Providing theoretical analysis to justify the approach and evaluating it across diverse data distributions and privacy budgets, with experimental results demonstrating its effectiveness in handling highly structured data heterogeneity in DPFL.

**Audience:**

Yes

**Broader Impact Concerns:**

NA.

**Claims And Evidence:**

Yes

**Requested Changes:**

1. Explains gradient accumulation for full batch training in plain language, and provides GMM-based clustering code snippets for reproducibility.
2. Add experiments on smaller datasets to quantify the trade-off between $b_i^{>1}$ reduction and computational overhead.

**Strengths And Weaknesses:**

Strengths:
1. The two-phase strategy (GMM soft clustering based on model update first, then loss-based clustering) addresses the limitations of existing loss-based or gradient-based methods.
2. Solid theoretical foundations link batch size selection with noise reduction, cluster separability, and GMM convergence, enhancing the effectiveness of the algorithm.
3. Experiments on different datasets, privacy budgets, and data partitions demonstrate its generalizability, with an explicit focus on performance on minority class clustering.

Weaknesses:
1. R-DPCFL relies on tuning batch sizes $\left(b_i^{>1}\right)$, strategy switching time ( $E_c$ ), and cluster count ( $M$ ). While guidelines exist, their optimal selection remains non-trivial for new datasets.
2. Performance degrades with small local datasets, requiring very small $b_i^{>1}$ for compensation, which may increase computational cost.

---

> ### Author Response · Authors · 2025-07-12
> **Official comment**
>
> We appreciate the reviewer for the constructive comments and questions. We are happy to answer them in the following:
>
> > Comment1: R-DPCFL relies on tuning batch sizes $b_i^{>1}$, strategy switching time $E_c$, and cluster count $M$. While guidelines exist, their optimal selection remains non-trivial for new datasets.
>
> We would like to remind that the only parameter that needs to be tuned as a hyper-parameter is the batch size $b_i^{>1}$, for which we have provided some guidelines to make its tuning easier. The switching time $E_c$ will be automatically set after learning the GMM using the formula incorporating $MPO$, explained at the end of page 11. Also, note that when $M$ is known, it is well-defined for all algorithms, including the baselines. However, when it is unknown, the IFCA baseline does not have any mechanism to find/estimate it, while R-DPCFL can find it exactly or at least estimate it closely. So this point is not only a weakness of R-DPCFL but also it can be counted as its strength/applicability in real scenarios.
>
> > Q1: Explain gradient accumulation for full batch training in plain language.
>
> We have explained gradient accumulation in Section G of the appendix in detail. In ordinary batch gradient descent, we set the gradient w.r.t every parameter to 0 at the beginning of each back propagation and then compute the gradient w.r.t each parameter. Finally, the gradients are used for updating the parameters at the end of the back propagation. In contrast to this, in gradient accumulation, gradients w.r.t parameters are summed over “multiple” ($\gamma$) smaller batches ($b$), followed by updating parameters  with the summation of the smaller batch gradients. In other words, in gradient accumulation, parameters are updated after every $\gamma$ back propagations with smaller batch sizes $b$. This simulates a batch size $\gamma \times b$ **with zero overheads**. We have added the above explanations to appendix G (in blue).
>
> > Q2: Provide GMM-based clustering code snippets for reproducibility.
>
> We will make our code publicly available and include it in the final version of our paper, for reproducibility by readers.
>
> > Q3: Add experiments on smaller datasets to quantify the trade-off between $b_i^{>1}$ reduction and computational overhead.
>
> DPSGD needs to compute and clip sample gradients in a batch one by one and use their average for parameter updating. Keeping this point in mind, **with a fixed dataset size**, using smaller batches is expected to increase the runtime to some extent, because with a fixed dataset size and a smaller batch size, it will take a longer time to complete a fixed number of training epochs. However, using a smaller batch size in order to compensate for **the smaller size of a client’s dataset** is expected to not affect the run time much, because the dataset size has also decreased in this case. On the other hand, using smaller batch sizes always reduces the memory overhead, if there are any constraints on that.
>
> We have included some results with smaller dataset sizes and smaller batch sizes in Fig. 9, on the right. Let us consider two cases in this figure as an example. When $N_i=10^4, b_i^{>1}=32, E=200, K=1$, each client performs $(200-1) * 1 * 10^4 / 32 = 62188$ gradient updates after the first round until the end. In contrast, when $N_i=6600$ (**i.e. a smaller dataset**), $b_i^{>1}=16$ (**i.e. 2 times smaller batch size**), $E=200, K=1$, each client performs $(200-1) * 1 * 6600 / 16 = 82088$ gradient updates after the first round until the end of training. **Also, each of the gradient updates are 2 times lighter in terms of their computational overhead (smaller batch size)**. So overall, the computational overhead does not even increase.

---

### Author Response · Authors · 2025-07-17
**Thank you all for the reviews**

Dear action editor and reviewers,

As the author-reviewer discussion deadline is approaching, we would like to take the chance to thank you all again. Our draft could not be improved without your constructive comments and suggestions, and we sincerely appreciate the time and effort you put in this draft. In the remaining time, we would be glad to answer any other questions that you may have. Thank you.

Paper 5060 Authors

---

### Decision · Action_Editor_Ros8 · 2025-08-28

**Recommendation:** Accept with minor revision

**Additional Comments:**

The author response addressed the key questions and concerns, and all three reviewers support acceptance.

I ask the authors to carefully incorporate the additional details and experiments provided in their response to the final manuscript.

**Audience:**

Yes

**Audience Explanation:**

The paper is about private federated learning, which is of broad interest to TMLR's audience.

**Claims And Evidence:**

Yes

**Claims Explanation:**

The paper provides both theoretical analysis and empirical results, offering strong evidence that the approach is well-founded and outperforms existing baselines for the problem.